# The Effects of Certificate-of-Need Laws on the Quality of Hospital Medical Services

## Thomas Stratmann

Economics Department, George Mason University, 4400 University Drive, MSN 1D3, Carow Hall, Fairfax, VA 22030, USA; tstratma@gmu.edu

**Abstract:** Certificate-of-need (CON) laws restrict entry into health services by requiring healthcare providers to seek approval from state healthcare regulators before making any major capital expenditures. An important question is whether CON laws influence the quality of medical services in CON law states. For instance, if CON laws actually lower the quality of medical services, they fail to achieve their intended effect. This paper tests the hypothesis that hospitals in states with CON laws provide lower-quality services than hospitals in states without CON laws. Our overall results suggest that CON regulations lead to lower-quality care for some quality measures and have little or no effect on other quality standards. The results remain consistent across several robustness tests.

**Keywords:** certificate-of-need; quality of medical services; health regulations

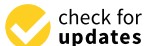



## 1. Introduction

Since the mid-1970s, most states have required healthcare providers to seek approval from the relevant state's healthcare planning agency before making significant capital expenditures. Today, the laws of 36 states and the District of Columbia allow state regulators to approve or reject spending on new facilities, devices, and services based on community "need". These certificate-of-need laws, or CON laws, aim to restrain healthcare spending.[1]

The objective of CON laws is to limit entry into the medical profession (Polsky et al. 2014; Baker and Stratmann 2021). As such, CON laws restrict new providers from entering the marketplace, thereby reducing competitive pressures for incumbent providers (Fayissa et al. 2020). Hospitals compete, *inter alia*, on quality of medical services; as such, we might expect hospitals facing fewer competitive pressures to see a drop in quality of patient services.

Hospitals also compete on the price of medical services. However, they have limited ability to compete on this front, as hospital reimbursement through insurers, tends to be determined administratively, rather than through market forces. As a result, hospitals mighthave an incentive to compete more intensely on non-price margins such as quality of medical services (Li and Dor 2015)

Economic theory predicts that when hospitals face regulated prices, free entry and competition will increase the equilibrium quality of patient care. By contrast, hospitals facing market-determined prices may compete on price and quality margins. The effect of free entry and competition on equilibrium hospital quality in a system of market-determined prices is therefore ambiguous. Gaynor and Town (2011, pp. 81–82) in their review of the literature on competition in healthcare markets, found that empirical work generally confirms these theoretical predictions: "Most of the studies of Medicare patients show a positive impact of competition on quality", whereas "the results from studies of markets where firms set prices (e.g., privately insured patients) are much more variable".

Supporters of CON regulations suggest that these regulations positively impact healthcare quality. For instance, the American Health Planning Association (AHPA) responding to a Federal Trade Commission critique of CON laws, argued that "recent empirical evidence shows substantial economic and service quality benefits from CON regulation and

related planning" (AHPA (American Health Planning Association) 2005, p. 14). Further, Thomas Piper, director of Missouri's CON program, told a joint Federal Trade Commission–Department of Justice hearing that "quality is improved" thanks to Missouri's CON program (Piper 2003, p. 27).

Specifically, CON supporters argue that a state regulator's ability to set standards and monitor utilization rates positively affects the quality of healthcare services (Thomas 2015; Steen 2016). This argument derives from research linking procedural volume with better outcomes: as practitioners repeatedly serve patients with the same conditions and repeatedly perform the same procedure, they become more specialized and proficient, leading to better patient outcomes (Halm et al. 2002). Using CON laws to restrict the number o fproviders helps regulators allocate more patients to existing providers, thereby increasing provider expertise and improving patient outcomes (Cimasi 2005).

Still, several scholarly works find no evidence of a systematic difference in the quality of care between providers in states governed by CON laws and those in non-CON states. For example, Polsky et al. (2014) examined the effects of CON laws on home healthcare services and found no significant differences in rehospitalization rates or expenditures between CON and non-CON states. Further, Paul et al. (2014) found that CON laws are associated with shorter emergency department visits. Likewise, Lorch et al. (2011) found that low-birth-weight mortality rates do not vary significantly between CON and non-CON states. However, CON states with large metropolitan areas have lower all-infant mortality rates than non-CON states. Related studies include

Several studies have found contradictory results regarding the relationship between CON laws and mortality after coronary artery bypass graft (CABG) surgeries. Cutler et al. (2009) found that CABG mortality rates declined after Pennsylvania repealed its CON laws. However, Ho et al. (2009) found no difference in CABG mortality rates between CON and non-CON states. Two studies of 1990s data have reported that higher CABG mortality rates in non-CON states (Vaughan-Sarrazin et al. 2002; Rosenthal and Sarrazin 2001). Related studies include (Browne et al. 2018; Casp et al. 2019; Ettner et al. 2020; Herb et al. 2021; Ohsfeldt and Li 2018; Rahman et al. 2016; Schultz et al. 2021; Stratmann and Wille 2016; Wu et al. 2019; Yuce et al. 2020; Ziino et al. 2021).

Studies examining the effect of CON laws on healthcare quality typically suffer from two limitations. First, inadequate data on provider quality constrains investigation into how CON regulations affect the quality of specific procedures, such as CABG, rather than considering quality across multiple margins. Second, studies on this topic, other than Polsky et al. (2014), struggle to untangle the causal effect of CON laws from other essential factors that independently affect healthcare quality and that might be correlated with whether a state has a CON program.

This paper proposes an empirical design that addresses those omitted-variable issues and allows us to estimate a causal effect. First, we exploit a dataset whose stated purpose is to measure hospital quality objectively across many aspects of the patient experience. Second, we build on the identification strategy of Polsky et al. (2014), allowing us to estimate the causal effect of CON regulations on the quality of hospital services. This empirical strategy compares outcomes of hospitals in a particular healthcare market in a CON state with those in the same healthcare market in a non-CON state. By focusing only on hospitals in these specific markets and assuming that unobserved patient- and geographic-level heterogeneities are similar on both sides of the CON border within one market, we can estimate the causal effect of CON regulations on hospital quality.[2]

The data used in our analysis come from Hospital Compare, a database maintained by the Centers for Medicare and Medicaid Services (CMS). Hospital Compare contains more than 100 quality indicators from more than 4000 Medicare-certified hospitals (CMS (Centers for Medicare & Medicaid Services) 2016a). These measures include readmission and mortality rates for common conditions, quality- and process-of-care indicators, and patient-experience surveys. The reason CMS used these measures is that they represent some of the most common, costly, and variable factors affecting individual hospitals'

performance. When considered together, these measures help to provide a summary of a hospital's overall quality of care. According to CMS, these data are aim ed at providing consistent and objective tools for patients to compare quality when selecting a healthcare provider. We assess the effect of CIN laws on hospital quality using provider-level quality metrics for nine different conditions from more than 900 hospitals from 2011 to 2015.

Our findings show that the quality of hospital care in states with CON laws is not systematically superior to the corresponding quality in non-CON states. Moreover, we find support for the hypothesis that hospitals in CON states tend to provide lower-quality services. In particular, we find that mortality rates for pneumonia and heart failure are significantly higher in hospitals in CON states. We also find that deaths from complications after surgery are considerably higher in CON states. Further, our findings show that CON regulations are associated with lower overall hospital quality, although the corresponding point estimates are not always precise. We present balancing tests and conduct several robustness tests. The results support the causal interpretation of our findings.

## 2. Regulatory Background

CON programs took effect nationwide after the National Health Planning and Resources Development Act of 1974 became law (Cimasi 2005). The act comprised part of the federal government's plan to develop a national health planning policy. The legislation required federal agencies to establish specific health policy goals, priorities, and guidelines (Cimasi 2005). The act also incentivized all 50 states to adopt a process through which healthcare providers would seek approval from their state's health planning agency before making any significant capital expenditure, such as a building expansion or purchasing new medical devices (NCSL (National Conference of State Legislatures) 2016). The stated goal of this policy was to ensure that the proposed additional medical services did not exceed community needs. Once a regulator determined a community need, the applicant was granted permission to commence the project, hence the term *certificate of need* (NCSL (National Conference of State Legislatures) 2016).

The National Health Planning and Resources Development Act of 1974 provided strong incentives to the 50 states to implement CON programs. The incentives tied specific federal healthcare funding to states' enacting CON programs (Cimasi 2005). Additinoally, the federal government directly subsidized the development of state CON programs. These national policies encouraged states without a CON program to adopt CON regulations. In 1974, 23 states had some form of CON regulations, and by 1980 the number had increased to 49. The National Health Planning and Resources Development Act of 1974 was repealed in 1986, lifting the requirement for states to maintain CON programs. The Act also eliminated the associated federal subsidy. Subsequently, some states maintained their CON laws, and others repealed them. Figure 1 lists the states with CON laws for 2011–2015 and the facilities, equipment, and procedures those states regulated. States did not significantly change their CON programs between 2011 and 2015.

In states with CON programs, healthcare providers seeking to enter a market, expand their facilities, or offer new services must apply to their state's healthcare planning agency for approval. Virginia, a state with a CON program covering comparatively many aspects of medical care, is representative in this regard.[3] Applicants must first submit a letter of intent to the Virginia Department of Health and the appropriate regional health planning agency. Next, the applicant must submit a formal application and pay a fee of up to USD 20,000. State regulators review submissions in 60-day batches, depending on the type of facility or procedure under review. The Code of Virginia requires regional healthcare planning agencies to hold at least one public hearing for each application. At this point, competitors of an applicant are allowed to challenge the need for the proposed medical service. Regional planners then submit their recommendations and reasoning to the department, which reviews the applications and proposals and may hold additional hearings. At the end of this process, the department makes a recommendation, which is sent to the state health commissioner for final approval or denial (Virginia Department of Health 2015).

| | Acute Hospital Beds | Air Ambulance | Ambulatory Surgical Centers | Burn Care | Cardiac Catheterization | Computed Tomography Scanners | Gamma Knives | Home Health | Hospice | Intermediate Care Facilities/Mental Retardation | Long Term Acute Care | Lithotripsy | Nursing Home Beds/Long Term Care Beds | Medical Office Buildings | Mobile Hi Technology | Magnetic Resonance Imaging (Scanners) | Neo-Natal Intensive Care | Obstetrics Services | Open Heart Surgery | Organ Transplants | Positron Emission Tomography Scanners | Psychiatric Services | Radiation Therapy | Rehabilitation | Renal Failure/Dialysis | Assisted Living & Residential Care Facilities | Subacute Services | Substance/Drug Abuse | Swing Beds | Ultra-Sound | Counts by State |
|------|---|---|---|---|---|---|---|---|---|---|---|---|---|---|---|---|---|---|---|---|---|---|---|---|---|---|---|---|---|---|----|
| AK | X | | X | | X | X | X | | | | X | X | X | | X | X | X | X | X | X | X | X | X | | X | | | X | | | 19 |
| AL | X | X | X | X | X | | X | X | X | | X | | X | | | X | X | X | X | | X | X | X | X | X | | | | X | X | 20 |
| AR | | | | | | | | X | X | X | | | X | | | | | | | | | | X | | | | | X | | | 6 |
| CT | X | | X | | X | X | | | X | | | X | | X | X | X | X | X | X | X | X | | | | | | | X | | | 17 |
| DC | X | X | X | X | X | X | X | X | X | | X | X | X | X | X | X | X | X | X | | X | X | X | X | X | | | X | X | X | 28 |
| DE | X | | X | | X | | | | | | X | X | X | | | | | | X | | | | X | | X | | | | | | 8 |
| FL | X | | | | | | | X | X | X | | | X | | | | | X | | | X | | X | | X | | | X | X | | 11 |
| GA | X | | X | | X | | X | X | | X | X | | X | | | X | X | X | X | | X | X | X | | | | | X | | | 17 |
| HI | X | | X | X | X | X | X | X | X | X | X | X | X | | X | X | X | X | X | X | X | X | X | X | X | | X | X | X | X | 27 |
| IA | X | | X | | X | | | | | X | X | | X | | | | X | X | X | | X | X | X | | | | | | | | 10 |
| IL | X | | X | | X | | | | | X | X | | X | | | X | X | X | X | | X | X | | | X | | X | | X | | 14 |
| KY | X | | X | | X | | | X | X | X | | | X | | X | X | X | X | X | X | X | X | X | | | | | X | | | 18 |
| LA | | | | | | | | | | X | | | X | | | | | | | | | | | | X | | | | | | 3 |
| MA | | X | X | | | | X | | | | X | X | | X | X | X | X | X | X | X | X | | | | | | X | | | | 14 |
| MD | X | | X | X | X | | | X | X | X | X | | X | | | X | X | X | X | X | X | | X | | X | | | | X | | 16 |
| ME | | X | X | X | X | X | X | | | | X | X | X | | X | X | X | X | X | X | X | X | X | X | | | X | X | X | X | 23 |
| MI | X | X | X | | X | X | X | | | X | X | X | X | | X | X | X | X | X | X | X | | | | | | | X | | | 18 |
| MO | X | | | | X | X | X | | | X | X | X | X | | X | X | | | X | | X | X | X | X | | X | | | | | 14 |
| MS | X | | X | | X | | X | X | X | X | | | X | | X | | | X | | X | X | X | X | X | | | X | | X | X | 18 |
| MT | | X | | | | | | X | X | | | | X | | | | | | | X | | | | | X | | | X | X | | 7 |
| NC | X | | X | X | X | X | X | X | X | X | X | X | X | | X | X | X | | X | X | X | X | X | X | X | X | X | X | | | 25 |
| NE | | | | | | | | | | | | | X | | | | | | | | | | | | X | | | | | | 2 |
| NH | | | X | | X | | | | | X | | | X | | X | X | | X | | X | X | X | | | X | | | X | | | 12 |
| NJ | X | | | X | X | | | X | | X | X | | X | | | X | | X | X | | X | | | | X | | | X | | | 12 |
| NV | X | | X | | | | | | | X | | | X | | | | | | | | | | | | | | | | | | 4 |
| NY | X | | X | X | X | X | | X | X | | X | X | X | X | X | X | X | | | | X | X | X | | | | | | | | 18 |
| OH | | | | | | | | | | | | | X | | | | | | | | | | | | | | | | | | 1 |
| OK | | | | | | | | X | | | X | | | | | | | | | | X | | | | | X | | | | | 4 |
| OR | | | | | | | X | | | X | | X | X | | | | | | | | | | | | X | | | | X | | 4 |
| RI | X | | X | | X | X | X | | X | | | X | | X | X | X | X | X | X | X | X | X | X | | | | | X | X | | 20 |
| SC | X | | X | | X | X | X | X | X | X | X | X | X | | X | X | X | | X | | X | X | X | X | | | | X | X | | 20 |
| TN | X | | X | X | X | | | X | X | X | X | X | X | | | X | X | | X | | X | X | X | X | | | | X | X | X | 20 |
| VA | X | | X | | X | X | X | | | X | X | X | X | | X | X | X | X | X | X | X | X | X | X | X | | | | | | 19 |
| VT | X | X | X | X | X | X | X | X | X | X | X | X | X | X | X | X | X | X | X | X | X | X | X | X | X | X | X | X | X | X | 30 |
| WA | X | | X | X | X | | | X | X | | X | | X | | | | X | X | X | X | X | | X | | X | X | | X | | X | 17 |
| WI | | | | | | | | | | X | | | X | | | | | | | | | | | | | | | X | | | 3 |
| WV | X | | X | | X | X | | X | X | X | X | | X | | X | X | X | X | X | X | X | X | X | X | X | X | | X | | | 21 |

**Figure 1.** Certificate-of-need regulations in the United States. Source: National Conference of State Legislatures (NCSL (National Conference of State Legislatures) 2016).

The criteria for assessing CON applications are usually specified in regulations promulgated by each state's planning agency (Cimasi 2005). For instance, Virginia mandates that the state health commissioner consider eight factors when assessing the public need for a new project: (1) whether the project will provide or increase access to health services; (2) whether the project will meet the needs of residents; (3) whether the project is consistent with current rules for medical facilities, such as minimum utilization rates; (4) to what extent the project will foster healthy competition; (5) how the project will affect the healthcare system, such as the utilization and efficiency of existing facilities; (6) the project's feasibility, including financial costs and benefits; (7) the extent to which the project will provide improvements in the financing and delivery of services; and (8) the project's contribution to research, training, and improvements to health services, in the case of a project proposed by or affecting a teaching hospital (Va. Code § 32.1-102.3 (2009)). However, the Code of Virginia does not rank these criteria concerning by their importance, leaving regulators discretion to weigh each criterion.

State CON program no only monitor and manage applications for proposed healthcare projects, but also set standards governing the use of facilities and procedures (Cimasi 2005). Virginia's CON program sets rules applying to 18 different healthcare services and facilities; collectively, these rules are called the State Medical Facilities Plan. For example, in the section that sets standards for CT scans, the plan states that "CT services should be within 30 min driving time one way under normal conditions of 95% of the population of the health planning district" (12 Va. Admin. Code § 5-230-90 (2009)). Other aspects of the plan set standards for determining minimum utilization rates, the timing for services to be introduced or expanded, staffing levels, and the minimum number of bassinets at facilities offering newborn services.

This level of specificity is typical for state CON programs. The South Carolina Health Plan, for example, requires applicants seeking a CON for diagnostic catheterization services to "project that the proposed service will perform a minimum of 500 diagnostic equivalent procedures annually within three years of initiation of services, without reducing the utilization of the existing diagnostic catheterization services in the service area below 500" per laboratory (South Carolina Health Planning Committee 2015, VIII-5). Similarly, Missouri's CON regulations state that "approval of additional intermediate care facility/skilled nursing facility (ICF/SNF) beds will be based on a service area need determined to be fifty-three (53) beds per one thousand (1000) population age sixty-five (65) and older minus the current supply of ICF/SNF beds" (Mo. Code Regs. tit. 19, § 60-50.450 (2014)).

## 3. Data

We use CMS metrics to estimate the difference in quality between hospitals in CON and non-CON states, including rates at which patients develop or die from surgical complications, patient survey results, readmission rates, and mortality rates. Here, we explain where and how we obtained those data and why we chose those specific metrics. This section also describes the aspect of quality each metric is intended to capture, how each metric is calculated, and our reasoning for including these metrics for measuring hospital quality.

This study analyzes CMS Hospital Compare data. Hospital Compare was launched in 2005 to "make it easier for consumers to make informed healthcare decisions and support efforts to improve quality in U.S. hospitals" (CMS (Centers for Medicare & Medicaid Services) 2016a). CMS partners with the Hospital Quality Alliance, whose members include the American Hospital Association, American Medical Association, and U.S. Chamber of Commerce. Before Hospital Compare, hospitals reported quality measures voluntarily. The Medicare Modernization Act of 2003 included incentives for hospitals to begin reporting data to CMS (Werner and Bradlow 2006). Today, CMS requires hospitals seeking reimbursement for any services funded by Medicare or Medicaid to provide data about the quality of their services and meet minimum quality thresholds (Medicare.gov 2016).



For the years 2011–2015, we analyze the effect of state CON laws on nine different quality-of-care indicators.

One measure meant to capture the quality of surgical patient care is *Deaths among Surgical Inpatients with Serious Treatable Complications* (PSI #4). This measure is a composite of mortality rates. It measures how many deaths occur per 1000 patients who develop a severe complication after surgery. Hospital Compare considers this measure an indicator of quality, as higher-quality hospitals identify complications sooner, treat them correctly, and thus incur fewer patient deaths.[4]

The denominator in PSI #4 comprises all hospital-level surgical discharges age 18 and older who developed care complications, including pneumonia, pulmonary embolism or deep vein thrombosis, sepsis, shock or cardiac arrest, and gastrointestinal hemorrhage or acute bleeding or acute ulcer. The numerator in PSI #4 comprises all discharged patients (included in the denominator) who died after developing a complication. Excluded this metric are patients aged 90 and older, patients transferred to an acute-care facility, and patients with missing discharge disposition, gender, age, quarter, year, or principal diagnosis information. The annual rate for the *Death among Surgical Inpatients with Serious Treatable Complications* measure is calculated using data over 20 months. For example, the data used to compute this measure in 2011 are from October 2008 to June 2010.

*Postoperative Pulmonary Embolism or Deep Vein Thrombosis* (PSI #12) measures the number of cases of pulmonary embolism or deep vein thrombosis per 1000 adult surgical discharges. According to the Centers for Disease Control and Prevention (CDC), patients recovering from surgery face an increased risk of developing potentially deadly blood clots in their deep veins (deep vein thrombosis) and lungs (pulmonary embolism) (CDC (Centers for Disease Control and Prevention) 2016). Page (2010) notes that a 2010 study by the Healthcare Management Council found that postoperative pulmonary embolism and deep vein thrombosis were the second most common hospital-acquired conditions after bedsores. These conditions are also the most expensive to treat, averaging USD 15,000 per case, or USD 564,000 per hospital annually. The denominator of this metric comprises all patients aged 18 and older who underwent an operating-room procedure. The numerator comprises all patients included in the denominator who developed deep vein thrombosis or pulmonary embolism as a secondary diagnosis. Excluded were patients diagnosed with deep vein thrombosis or pulmonary embolism before or on the same day as the first operating-room procedure, patients undergoing childbirth, and patients with missing discharge disposition, gender, age, quarter, year, or principal diagnosis information. The annual rate for the *Postoperative Pulmonary Embolism or Deep Vein Thrombosis* measure is calculated using data collected over 20 months. For example, the data used to compute this measure in 2011 were collected from October 2008 to June 2010.

Another hospital quality measure, which comes from the Hospital Consumer Assessment of Healthcare Providers and Systems (HCAHPS) survey, is the percentage of patients surveyed who rate their hospital a 9 or 10 overall during their last inpatient stay, on a scale of 1 (lowest) to 10 (highest). The survey was developed in 2005 by CMS in partnership with the Agency for Healthcare Research and Quality (CMS (Centers for Medicare & Medicaid Services) 2016c). This survey is based on a standardized instrument and data collection methodology that allows for cross-hospital comparisons of patients' experiences with different aspects of care. The instrument contains 27 questions, including one asking patients to provide an overall rating of their hospital on a 10-point scale. CMS segments the survey data into three tranches: the percentage of patients who rated their hospital as "low", defined as 6 or below; the percentage of patients who rated their hospital as "medium", defined as seven or eight; and the percentage of patients who rated their hospital as "high", defined as 9 or 10. We use the final measure in our analysis.

The HCAHPS survey is administered to a random sample of eligible hospital patients, including all inpatients who did not receive a psychiatric diagnosis. Excluded from the sample are patients in hospice and nursing home care, prisoners, patients with foreign home addresses, and patients excluded due to local regulations. Hospitals survey their

eligible sample of patients randomly each month, and are required to complete at least 300 surveys over 12 months. Patients in the sample are surveyed 48 h to six weeks after the discharge. Hospital-level results are updated on the Hospital Compare website every quarter, and each quarter's measures are based on the previous 12 months of data. CMS adjusts the HCAHPS data based on each hospital's patient mix. This adjustment allows for comparisons across hospitals with heterogeneous patients.[5]

We include six additional hospital quality variables: Pneumonia Readmission Rate (READM-30-PN), Pneumonia Mortality Rate (MORT-30-PN), Heart Failure Readmission Rate (READM-30-HF), Heart Failure Mortality Rate (MORT-30-HF), Heart Attack Readmission Rate (READM-30-AMI), and Heart Attack Mortality Rate (MORT-30-AMI). These variables measure the readmission and mortality rates for pneumonia, heart failure, and heart attack patients separately. These measures represent conditions with high morbidity and mortality rates that "impose a substantial burden on patients and the healthcare system" and for which "there is marked variation in outcomes by institution" (CMS (Centers for Medicare & Medicaid Services) 2012, p. 3). Moreover, these metrics are commonly used to evaluate hospital quality (Werner and Bradlow 2006; Zuckerman et al. 2016).

Readmission rates measure unplanned readmissions for any cause to an acute-care hospital within 30 days of discharge from a hospitalization for the given medical condition. Mortality rates measure deaths for any reason within 30 days of hospital admission for patients hospitalized with the given medical condition. CMS computes the readmission and mortality rates using a hierarchical model and then "risk standardizes" these measures. Thus, these rates consider patient characteristics that may make death or unplanned readmission more likely. Further, these rates account for hospital-specific effects: CMS estimates are based on a specific hospital's impact on its patients' likelihood of being readmitted or dying.

A hospital's ris—standardized readmission rate and risk-standardized mortality rate constitue the ratio of predicted readmissions or deaths associated with a given condition to the number of expected readmissions or deaths related to that condition. The predicted rate estimates the number of readmissions or deaths within 30 days at a given hospital for patients discharged for a given condition. This rate considers the hospital's patient risk factors (estimated from hospital-specific patient administrative data collected by CMS) and includes an estimate of the hospital-specific effect.[6]

The risk-standardized readmission rate and the risk-standardized mortality rate comprise patients who are Medicare fee-for-service beneficiaries aged 65 and older discharged from nonfederal acute-care hospitals with a principal discharge diagnosis of pneumonia, heart failure, or heart attack. The measures exclude admissions for patients discharged on the day of admission or the following day, those transferred to another acute care hospital, those enrolled in a Medicare hospice program any time in the 12 months before the hospitalization, those discharged against medical advice, and those who were not previously hospitalized in the 30 days before death. The data for the annual risk-standardized readmission rate and the risk-standardized mortality rate were collected over three years. This approach increases the number of cases per hospital, which allows for a more precise estimate and thus accommodates greater variation in hospital performance (CMS (Centers for Medicare & Medicaid Services) 2007). For example, the measures for 2011 use data collected from July 2007 to June 2010.

CMS collects Hospital Compare data and recalculates the quality measures periodically, usually annually or quarterly. CMS updates the measures analyzed in this study annually. Hospital Compare data might be missing for any given provider for several reasons. There might be too few cases or patients to report data for a given condition because the number does not meet the minimum threshold for public reporting. In such cases the number of patients is too small to generate a reliable estimates. CMS might not include provider data due to data inaccuracies, because a hospital does not have data that meet the selection criteria, or because no data are available.

Due to variations in data availability, the number of providers differs by the type of quality measure. Some hospitals have no reported data for some measures. Missing data can be a potential drawback of our identification strategy because a hospital's decision about whether to report data may be nonrandom (Werner and Bradlow 2006). For example, missing data might be correlated with lower quality. If so, and if CON laws are indeed associated with lower-quality hospitals, then we would underestimate any adverse effect, in absolute value, of CON laws on quality.

More aggregate hospital quality measures became available only recently. As part of the Dartmouth Atlas of Health Care (Dartmouth Atlas Project 2016), several aggregate quality measures were constructed to capture the rate of posthospitalization events among Medicare patients. In 2013, Hospital Compare began providing aggregate quality indicators to facilitate high-level hospital comparisons. In addition to medical-condition-specific quality measures, we test whether these aggregate quality measures differ between hospitals in CON and non-CON states.

## 4. Empirical Framework

Our identification strategy exploits the fact that, on occasion, a local healthcare market is divided between two states, one with a CON law and the other without. Our measure for a local healthcare market is a hospital referral region (HRR), which comes from the Dartmouth Atlas of Health Care (Dartmouth Atlas Project 2016). HRRs are defined based on referral patterns for major cardiovascular surgical procedures and neurosurgery patients. There are 306 HRRs in the United States.

Our empirical model is

$$\text{Quality}_{ijm} = \beta_0 + \beta_1 \, CON_j + \boldsymbol{\beta_2} \, \boldsymbol{X}_{ijm} + \boldsymbol{v_m} + \varepsilon_{ijm}, \tag{1}$$

The dependent variable is a quality measure for hospital $i$ in state $j$ and healthcare market $m$. Thus, two or more states can be contained in each market. The variable $CON_j$ equals one if state $j$ has a CON law and zero otherwise.[7] The model also includes market-level fixed effects ($\boldsymbol{v_m}$). This model estimates the coefficient of interest, $\beta_1$, based on states that vary in whether they have a CON law and are in the same healthcare market.

Following Polsky et al. (2014, p. 3), we use the Dartmouth Atlas of Health Care's HRR as the identifying healthcare market because it "defines a contiguous locality within which most tertiary hospital care referrals are contained and because it is the area most linked to geographic variation". By estimating the coefficient on $CON_j$, we control for unobservable heterogeneity, such as geographic variation and illness severity, which varies across HRRs. The applicability of this model assumes that the markets that cross the borders of CON and non-CON states are otherwise the same. We test this assumption below.

Our empirical model also controls for demographic factors that may vary across CON and non-CON states and are also determinants of hospital quality. Specifically, we control for the percentage of people over age 65 in provider $i$'s county, the percentage of African American people, the percentage of Hispanic people, the percentage of people living in a rural communities, the percentage of the ninth-grade cohort graduating in four years, the percentage of people without insurance, the median income for individuals in provider $i$'s county, and the county unemployment rate. For example, hospitals in higher-income areas may appear to perform better on the quality metrics because wealthier patients may be healthier, on average, than less affluent patients. Similarly, hospitals in areas with a larger population of elderly residents may appear to provide worse-quality care because older people may be less healthy than younger people on average. All covariates are contained in the $\boldsymbol{X}$ vector in Equation (1).

In our preferred specification, we calculate the coefficient on CON in Equation (1) using a pooled panel regression with hospital-level quality data from 2011 to 2015. We cluster standard errors on two dimensions: the individual hospital level and the hospital referral region level. This addresses the fact that we observe the same hospital over multiple

years. We calculate the same equation for each year as a check, omitting the year dummy variables. In these specifications, we cluster standard errors at the state level to compensate for the fact the observations are not independent. Further, we offer additional robustness checks to determine whether chance findings drive our results.

Table 1 shows the reporting rates for our quality measures for the example year 2011, which has a reporting rate typical for the remaining years in our sample. In the total sample of 4542 hospitals, between 40 and 90 percent of hospitals reported data for a given measure. The lowest reporting rate is for *Death among Surgical Inpatients with Serious Treatable Complications* (42 percent). The highest reporting rate is for *Pneumonia Readmission Rate* and *Pneumonia Mortality Rate* (90 percent each). In our subsample, which consists of the 921 hospitals included in our empirical model for 2011, the reporting rate is slightly lower. Specifically, between 30 and 85 percent of hospitals reported data for a given measure. In this subsample, the reporting rates mirrored those from the overall sample. The lowest reporting rate was for *Death among Surgical Inpatients with Serious Treatable Complications* (32 percent). The highest reporting rate was for *Pneumonia Readmission Rate* and *Pneumonia Mortality Rate* (86 and 85 percent, respectively).

**Table 1.** Reporting rates for hospital quality metrics in the full sample and the restricted sample (2011).

| Measure Name (CMS Code) | Full Sample (*n* = 4538) | | | Restricted Sample (*n* = 921) | | |
|---|---|---|---|---|---|---|
| | Providers in CON States | Providers in Non-CON States | Overall Reporting Rate | Providers in CON States | Providers in Non-CON States | Overall Reporting Rate |
| Death among Surgical Inpatients with Serious Treatable Complications (PSI #4) | 1295 (44%) | 624 (39%) | 42% | 122 (30%) | 175 (34%) | 32% |
| Postoperative Pulmonary Embolism or Deep Vein Thrombosis (PSI #12) | 2013 (68%) | 1107 (70%) | 69% | 202 (50%) | 334 (64%) | 58% |
| Percentage of patients giving their hospital a 9 or 10 overall rating (HCAHPS) | 2498 (85%) | 1324 (83%) | 84% | 286 (71%) | 428 (83%) | 78% |
| Pneumonia Readmission Rate (READM-30-PN) | 2734 (93%) | 1348 (85%) | 90% | 364 (90%) | 425 (82%) | 86% |
| Pneumonia Mortality Rate (MORT-30-PN) | 2724 (92%) | 1339 (84%) | 90% | 361 (90%) | 423 (82%) | 85% |
| Heart Failure Readmission Rate (READM-30-HF) | 2637 (89%) | 1268 (80%) | 86% | 329 (82%) | 391 (75%) | 78% |
| Heart Failure Mortality Rate (MORT-30-HF) | 2602 (88%) | 1237 (78%) | 85% | 321 (80%) | 384 (74%) | 77% |
| Heart Attack Readmission Rate (READM-30-AMI) | 1602 (54%) | 725 (46%) | 51% | 145 (36%) | 216 (42%) | 39% |
| Heart Attack Mortality Rate (MORT-30-AMI) | 1866 (63%) | 838 (53%) | 60% | 172 (43%) | 253 (49%) | 46% |

Note: CMS = Centers for Medicare and Medicaid Services; CON = certificate of need; HCAHPS = Hospital Consumer Assessment of Healthcare Providers and Systems. The restricted sample reflects our fixed-effects model and includes only providers in HRRs that cross the borders between states with and without CON laws. Sources: CMS (Centers for Medicare & Medicaid Services) (2016d); Hospital Compare Data Archive (n.d.) (2011); Dartmouth Atlas Project (2016); NCSL (National Conference of State Legislatures) (2016).

Data on CON laws in each state are available from the National Conference of State Legislatures (NCSL (National Conference of State Legislatures) 2016). The number and type of medical devices and procedures regulated by CON laws vary across states. For example, the District of Columbia has extensive CON legislation, while Ohio only regulates nursing home beds. We define a state as having a CON law with at least one CON regulation. Since none of the 50 states or the District of Columbia changed their CON regulations between 2011 and 2015, state coding remained consistent over our entire sample.

The annual data for the control variables of income, age, education, unemployment, uninsured, and demographics are from the County Health Rankings website (County Health Rankings and Roadmaps 2017). We compiled our control variables at the county level using the County Health Rankings and Roadmap datasets.

## 5. Results

### 5.1. Descriptive Statistics

We began with a dataset with an average of 4630 hospitals per year from 2011 to 2015, for 23,152 observations. Of these hospitals, an average of 2989 per year are in the District of Columbia and the 36 states with some CON regulations between 2011 and 2015. On average, 1641 hospitals per year are in non-CON states.

Next, among all 306 HRRs in the country, we identified 39 HRRs containing hospitals in both CON and non-CON states each year except 2014. For 2014, we identified 38 HRRs that included CON and non-CON states. Figure 2 presents a map of the state-border-crossing HRRs in the contiguous United States. Table 2 provides a list of these HRRs and the CON and non-CON states located in each HRR for the year 2011. Table 3 shows the number of providers on the CON side and the non-CON side of the border in each HRR. The state-border-crossing HRRs contain, on average, 962 hospitals per year, of which 422 are in CON states and 540 in non-CON states. This subsample represents about 21 percent of the observations in our original dataset.

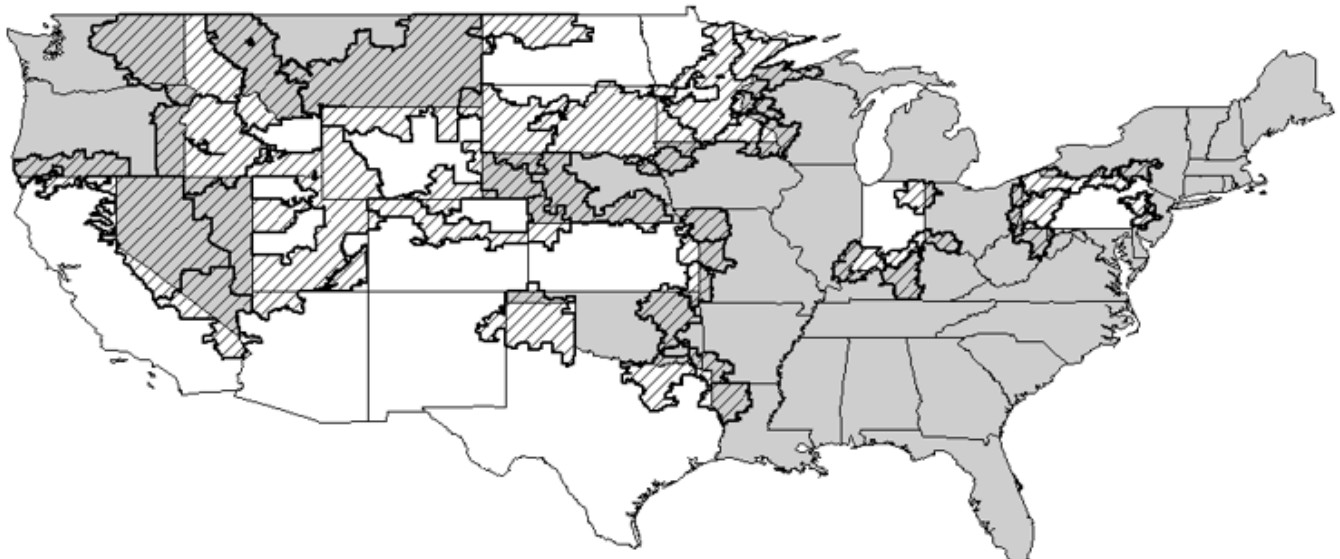

**Figure 2.** HRRs that crossed the borders of states with and without CON laws (2011). Note: HRRs = hospital referral regions; CON = certificate of need. Sources: NCSL (National Conference of State Legislatures) (2016); Dartmouth Atlas Project (2016).

**Table 2.** HRRs that crossed the borders of states with and without CON laws (2011).

| HRR Number | Non-CON States | CON States | HRR Number | Non-CON States | CON States |
|---|---|---|---|---|---|
| 22 | TX | AR, OK | 296 | PA | NY |
| 103 | CO, KS | NE | 324 | ND | MT |
| 104 | CO, WY | NE | 327 | IN | OH |
| 151 | ID | OR | 335 | PA | OH |
| 179 | IN | IL, KY | 340 | KS | OK |
| 180 | IN | OH | 343 | CA | OR |
| 196 | SD | IA, NE | 346 | PA | NJ |
| 205 | IN | KY | 351 | PA | NY, OH |
| 219 | TX | LA | 356 | PA | NJ |
| 250 | MN | MI, WI | 357 | PA | OH, WV |
| 251 | MN | WI | 359 | PA | NY |
| 253 | MN | IA | 370 | SD | NE |
| 256 | MN | WI | 371 | MN, SD | IA |
| 267 | KS | MO, OK | 383 | KS, NM, TX | OK |
| 268 | KS | MO | 391 | TX | OK |
| 274 | WY | MT | 423 | CO, ID, UT, WY | NV |
| 276 | ID | MT | 440 | ID | OR, WA |
| 277 | KS | NE | 445 | PA | MD, WV |
| 279 | AZ, CA | NV | 448 | MN | IA, WI |
| 280 | CA | NV | - | - | - |

Note: HRRs = hospital referral regions; CON = certificate of need. Sources: NCSL (National Conference of State Legislatures) (2016); Dartmouth Atlas Project (2016).

**Table 3.** Number of providers in HRRs that crossed the borders of states with and without CON laws (2011).

| HRR Number | Providers in Non-CON States | Providers in CON States | HRR Number | Providers in Non-CON States | Providers in CON States |
|---|---|---|---|---|---|
| 22 | 3 | 5 | 296 | 1 | 5 |
| 103 | 29 | 10 | 324 | 6 | 1 |
| 104 | 4 | 1 | 327 | 2 | 21 |
| 151 | 7 | 2 | 335 | 2 | 7 |
| 179 | 10 | 11 | 340 | 2 | 38 |
| 180 | 17 | 5 | 343 | 2 | 7 |
| 196 | 1 | 13 | 346 | 15 | 1 |
| 205 | 9 | 21 | 351 | 14 | 3 |
| 219 | 2 | 17 | 356 | 38 | 6 |
| 250 | 11 | 4 | 357 | 34 | 10 |
| 251 | 61 | 11 | 359 | 5 | 1 |
| 253 | 10 | 2 | 370 | 11 | 2 |
| 256 | 7 | 7 | 371 | 46 | 7 |
| 267 | 3 | 8 | 383 | 17 | 2 |
| 268 | 17 | 30 | 391 | 68 | 2 |
| 274 | 7 | 26 | 423 | 34 | 1 |
| 276 | 1 | 14 | 440 | 9 | 23 |
| 277 | 1 | 32 | 445 | 1 | 9 |
| 279 | 4 | 17 | 448 | 1 | 10 |
| 280 | 6 | 11 | - | - | - |

Note: HRRs = hospital referral regions; CON = certificate of need. Sources: NCSL (National Conference of State Legislatures) (2016); Dartmouth Atlas Project (2016).

Table 4, Panel A, shows whether there are any systematic differences in the population characteristics between states with and without CON regulations, using the year 2015 as an example. In each panel, the unit of observation is a hospital in 2015. Given that we know the location of each hospital, we match annual county-level variables to each hospital for each year that appears in our dataset. Panel A shows that counties in CON states tend to have a greater number of African Americans as a share of the total population than

counties in non-CON states. In comparison, non-CON states tend to have a greater number of Hispanics as a share of the total population. Counties in states with CON laws also tend to have higher unemployment rates than counties in non-CON states.

**Table 4.** Differences-in-means tests: covariates (2015).

| Panel A: All CON States versus All Non-CON States | Non-CON States | CON States | Difference | *t* Statistic | Observations |
|---|---|---|---|---|---|
| Percentage Over Age 65 | 15.6 | 15.8 | −0.3 | 0.42 | 51 |
| Percentage African American | 3.5 | 12.6 | −9.1 | 2.72 | 51 |
| Percentage Hispanic | 16.2 | 8 | 8.2 | −2.99 | 51 |
| Percentage Rural | 39.2 | 37.3 | 1.9 | −0.32 | 51 |
| Percentage Uninsured | 20.7 | 19.2 | 1.5 | −0.81 | 51 |
| Average Graduation Rate (HS) | 81.8 | 79.9 | 1.9 | −0.92 | 51 |
| Unemployment Rate | 6.2 | 7.4 | −1.2 | 2.36 | 51 |
| Household Income (USD) | 51,156 | 50,582 | 574 | −0.22 | 51 |
| **Panel B: HRRs in Both CON and Non-CON States** | **HRR in Non-CON** | **HRR in CON** | **Difference** | ***t* Statistic** | **Observations** |
| Percentage Over Age 65 | 16.7 | 17.2 | −0.5 | 0.78 | 78 |
| Percentage African American | 4.3 | 4.6 | −0.3 | 0.15 | 78 |
| Percentage Hispanic | 7.8 | 7.6 | 0.2 | −0.12 | 78 |
| Percentage Rural | 45.1 | 51.8 | −6.7 | 1.47 | 78 |
| Percentage Uninsured | 18.2 | 20.6 | −2.4 | 1.77 | 78 |
| Average Graduation Rate (HS) | 84.2 | 81 | 3.3 | −1.71 | 76 |
| Unemployment Rate | 6.5 | 6.8 | −0.4 | 0.83 | 78 |
| Household Income (USD) | 50,046 | 46,080 | 3965 | −2.12 | 78 |

Note: CON = certificate of need; HRRs = hospital referral regions. The unit of analysis is the individual provider. Data for percentage over age 65, percentage African American, percentage Hispanic, percentage rural, average freshman graduation rate (high school), percentage uninsured, median income, and unemployment rate are the average from the county level. Sources: CMS (Centers for Medicare & Medicaid Services) (2016d); Hospital Compare Data Archive (n.d.) (2015); Dartmouth Atlas Project (2016); American FactFinder (2016); NCSL (National Conference of State Legislatures) (2016); County Health Rankings and Roadmaps (County Health Rankings and Roadmaps 2011–2015).

Table 4, Panel B, shows results from testing whether there are any significant differences within the subsample of hospitals in HRRs that cross state borders, where state CON status varies within an HRR. The comparison of means differences shows statistically significant differences between the uninsured rate in CON and non-CON hospitals within a border-crossing HHR. CON hospitals within a border-crossing HHR tend to have a slightly higher uninsured rate. Moreover, residents of counties on the non-CON side of the border tend to be slightly better educated and to have slightly higher incomes than those on the CON side. Because higher levels of education and income may be associated with higher-quality hospitals, differences in these variables, to the extent they are unmeasured or not included in our regression model, will tend to overstate any adverse effect of CON laws on the quality of hospital services provided. Therefore, we include controls for both variables in our regression models.

### 5.2. Hospital Quality Indicators

Table 5 shows whether there are any significant quality differences between hospitals in CON states and non-CON states. We find that nearly all the quality measures are statistically significantly worse in CON states than in non-CON states. Among the nine quality-of-care measures, readmission rate, and mortality rate, only Heart Attack Mortality Rate is not significantly different between CON and non-CON states. The metrics with the largest-magnitude differences are *Pneumonia Readmission Rate* and *Heart Failure Readmission Rate*; on average, hospitals in CON states have over 0.5 percentage points more pneumonia and heart failure patient readmissions than non-CON states, implying about five additional readmissions per 1000 patient discharges.

**Table 5.** Differences-in-means tests: hospital quality indicators for all CON states vs. all non-CON states (2011–2015).

| Measure Name (CMS Code) | Mean Sample | Non-CON States | CON States | Difference | Clustered *t* Statistic | Observations |
|---|---|---|---|---|---|---|
| Death among Surgical Inpatients with Serious Treatable Complications (deaths per 1000 surgical discharges with complications) (PSI #4) | 115.1 | 113.2 | 116.0 | −2.9 | 4.24 | 9537 |
| Postoperative Pulmonary Embolism or Deep Vein Thrombosis (per 1000 surgical discharges) (PSI #12) | 4.5 | 4.3 | 4.6 | −0.3 | 4.95 | 15,390 |
| Percentage of patients giving their hospital a 9 or 10 overall rating (percentage points) (HCAHPS) | 69.7 | 70.5 | 69.3 | 1.2 | −4.35 | 19,853 |
| Pneumonia Readmission Rates (percentage points) (READM-30-PN) | 17.8 | 17.5 | 17.9 | −0.5 | 13.17 | 20,645 |
| Pneumonia Mortality Rate (percentage points) (MORT-30-PN) | 11.9 | 11.8 | 12.0 | −0.3 | 5.22 | 20,559 |
| Heart Failure Readmission Rate (percentage points) (READM-30-HF) | 23.5 | 23.2 | 23.6 | −0.5 | 9.79 | 19,316 |
| Heart Failure Mortality Rate (percentage points) (MORT-30-HF) | 11.7 | 11.6 | 11.8 | −0.2 | 3.79 | 18,901 |
| Heart Attack Readmission Rate (percentage points) (READM-30-AMI) | 18.5 | 18.3 | 18.7 | −0.4 | 8.20 | 11,377 |
| Heart Attack Mortality Rate (percentage points) (MORT-30-AMI) | 15.1 | 15.0 | 15.1 | −0.1 | 1.44 | 12,792 |

Note: CON = certificate of need; CMS = Centers for Medicare and Medicaid Services; HCAHPS = Hospital Consumer Assessment of Healthcare Providers and Systems. The unit of analysis is the individual hospital. Data are collected at the individual hospital level. Readmission and mortality rates are calculated using data from Medicare patients only. All *t* statistics are clustered at the individual provider level. Sources: CMS (Centers for Medicare & Medicaid Services) (2016d); Hospital Compare Data Archive (n.d.) (2011, 2012, 2013, 2014, 2015); Dartmouth Atlas Project (2016).

Table 6 provides results from studying differences in outcomes for the 39 HRRs that cross the border between a CON state and a non-CON state. Restricting the analysis to these HRRs, we again find statistically significant quality differences between hospitals on the CON side of the border and those on the non-CON side. *Heart Attack Mortality Rate* is the only metric that does not differ between CON and non-CON states. The largest-magnitude difference is *Pneumonia Mortality Rate*, and the magnitude of this estimate is similar to that shown in Table 5.

Table 6 shows that, for these 39 HRRS, hospitals in CON states appear to perform worse on all quality indicators. However, hospitals in CON states now perform better on average on *Postoperative Pulmonary Embolism* than hospitals in non-CON states, by about four cases per 1000 discharges. Nevertheless, these summary statistics of hospital quality indicators provide preliminary evidence that hospitals in states with CON regulations tend to score lower on quality measures than those without CON laws.



**Table 6.** Differences-in-means tests: hospital quality indicators for HRRs in both CON and non-CON states (2011–2015).

| Measure Name (CMS Code) | Mean Sample | HRRs in Non-CON States | HRRs in CON States | Difference | Clustered *t* Statistic | Observations |
|---|---|---|---|---|---|---|
| Death Among Surgical Inpatients with Serious Treatable Complications (deaths per 1000 surgical discharges with complications) (PSI #4) | 113.1 | 111.1 | 116.0 | −4.9 | 3.12 | 1539 |
| Postoperative Pulmonary Embolism or Deep Vein Thrombosis (per 1000 surgical discharges) (PSI #12) | 4.4 | 4.5 | 4.2 | 0.4 | −2.99 | 2779 |
| Percentage of patients giving their hospital a 9 or 10 overall rating (percentage points) (HCAHPS) | 71.3 | 71.9 | 70.5 | 1.4 | −2.37 | 4006 |
| Pneumonia Readmission Rate (percentage points) (READM-30-PN) | 17.6 | 17.5 | 17.7 | −0.2 | 2.83 | 4141 |
| Pneumonia Mortality Rate (percentage points) (MORT-30-PN) | 12.0 | 11.8 | 12.2 | −0.5 | 5.12 | 4112 |
| Heart Failure Readmission Rate (percentage points) (READM-30-HF) | 23.3 | 23.2 | 23.5 | −0.3 | 2.64 | 3659 |
| Heart Failure Mortality Rate (percentage points) (MORT-30-HF) | 11.8 | 11.6 | 12.1 | −0.4 | 4.87 | 3552 |
| Heart Attack Readmission Rate (percentage points) (READM-30-AMI) | 18.5 | 18.4 | 18.5 | −0.1 | 0.94 | 1806 |
| Heart Attack Mortality Rate (percentage points) (MORT-30-AMI) | 15.1 | 15.0 | 15.3 | −0.3 | 2.74 | 2033 |

Note: HRRs = hospital referral regions; CON = certificate of need; CMS = Centers for Medicare and Medicaid Services; HCAHPS = Hospital Consumer Assessment of Healthcare Providers and Systems. The unit of analysis is the individual hospital. Data are collected at the individual hospital level. Readmission and mortality rates are calculated using data from Medicare patients only. All *t* statistics are clustered at the individual provider level. Sources: CMS (Centers for Medicare & Medicaid Services) (2016d); Hospital Compare Data Archive (n.d.) (2011, 2012, 2013, 2014, 2015); Dartmouth Atlas Project (2016).

*5.3. Pooled Panel Regression Results*

Table 7 presents estimates from six regression models pooling annual data on hospital quality from 2011 to 2015. Here, we present only the coefficient of interest, the coefficient on the CON dummy variable in Equation (1). In each model specification, the CON coefficient is identified as the difference in the quality of medical services between hospitals in CON states and hospitals in non-CON states. Identification does not derive from variation over time in CON laws, as none of the 50 states or the District of Columbia changed their CON laws between 2011 and 2015.[8]

**Table 7.** Pooled regression results (2011–2015).

| Measure Name (CMS Code) | (A) Full-Sample Bivariate Model | (B) Full-Sample Multivariate Model | (C) Restricted-Sample Bivariate Model | (D) Restricted-Sample Multivariate Model | (E) HRR Fixed-Effects Bivariate Model | (F) HRR Fixed-Effects Multivariate Model |
|---|---|---|---|---|---|---|
| Death among Surgical Inpatients with Serious Treatable Complications (deaths per 1000 surgical discharges with complications) (PSI #4) | 2.861 *** (1.093) 9537 | 1.919 * (1.138) 9375 | 4.896 * (2.735) 1504 | 3.312 (2.425) 1490 | 6.011 *** (1.701) 1504 | 6.161 *** (2.278) 1490 |
| Postoperative Pulmonary Embolism or Deep Vein Thrombosis (per 1000 surgical discharges) (PSI #12) | 0.283 *** (0.0993) 15,390 | 0.248 ** (0.124) 14,771 | −0.434 ** (0.178) 2673 | −0.245 * (0.146) 2575 | −0.0643 (0.110) 2673 | 0.117 (0.150) 2575 |
| Percentage of patients giving their hospital a 9 or 10 overall rating (percentage points) (HCAHPS) | −1.163 * (0.616) 19,853 | −1.546 *** (0.508) 18,779 | −1.416 * (0.741) 3878 | −0.387 (0.682) 3633 | −1.334 (0.996) 3878 | −0.964 (0.957) 3633 |
| Pneumonia Readmission Rate (percentage points) (READM-30-PN) | 0.451 *** (0.0695) 20,645 | 0.369 *** (0.0696) 19,431 | 0.185 (0.126) 4082 | 0.146 * (0.0838) 3823 | 0.135 (0.0904) 4082 | 0.0854 (0.0958) 3823 |
| Pneumonia Mortality Rate (percentage points) (MORT-30-PN) | 0.258 ** (0.107) 20,559 | 0.0947 (0.0785) 19,362 | 0.486 *** (0.116) 4053 | 0.414 *** (0.0890) 3799 | 0.423 *** (0.135) 4053 | 0.379 *** (0.122) 3799 |
| Heart Failure Readmission Rate (percentage points) (READM-30-HF) | 0.481 *** (0.104) 19,316 | 0.464 *** (0.119) 18,344 | 0.268 (0.166) 3588 | 0.231 (0.147) 3427 | 0.291 * (0.163) 3588 | 0.248 (0.184) 3427 |
| Heart Failure Mortality Rate (percentage points) (MORT-30-HF) | 0.166 (0.129) 18,901 | 0.0608 (0.0883) 17,971 | 0.434 *** (0.146) 3486 | 0.344 *** (0.0923) 3338 | 0.248 ** (0.100) 3486 | 0.198 ** (0.0822) 3338 |
| Heart Attack Readmission Rate (percentage points) (READM-30-AMI) | 0.351 *** (0.0817) 11,377 | 0.351 *** (0.0819) 11,048 | 0.0984 (0.155) 1784 | 0.183 (0.121) 1747 | 0.139 (0.106) 1784 | 0.179 (0.124) 1747 |
| Heart Attack Mortality Rate (percentage points) (MORT-30-AMI) | 0.0706 (0.0939) 12,792 | −0.0245 (0.0722) 12,358 | 0.328 * (0.173) 2006 | 0.216 (0.137) 1956 | 0.321 * (0.170) 2006 | 0.263 (0.173) 1956 |
| Controls | No | Yes | No | Yes | No | Yes |
| Year | Yes | Yes | Yes | Yes | Yes | Yes |
| HRR fixed effects | No | No | No | No | Yes | Yes |

Note: CMS = Centers for Medicare and Medicaid Services; HRR = hospital referral region; HCAHPS = Hospital Consumer Assessment of Healthcare Providers and Systems. The model specifications in Columns A and B consider the full sample of hospitals in the United States. The specifications in Columns C through E consider only hospitals in HRRs that cross the border between CON and non-CON states. The unit of analysis is the individual provider. Clustered standard errors by provider and hospital referral region are in parentheses. Controls for percentage over age 65, percentage African American, percentage Hispanic, percentage rural, average freshman graduation rate (high school), percentage uninsured, median income, and unemployment rate are the average from the county level. Readmission and mortality rates are calculated using data from Medicare patients only. The number of observations varies between the bivariate and multivariate regressions and by measure. In each cell in the table, the top number is the coefficient estimate, the number in parentheses is the standard error, and the bottom number indicates the number of observations (details are available from the authors on request). *** Statistically significant at (at least) the 1% level. ** Statistically significant at (at least) the 5% level. * Statistically significant at (at least) the 10% level. Sources: CMS (Centers for Medicare & Medicaid Services) (2016d); Hospital Compare Data Archive (n.d.) (2011, 2012, 2013, 2014, 2015); Dartmouth Atlas Project (2016); American FactFinder (2016); County Health Rankings and Roadmaps (County Health Rankings and Roadmaps 2011–2015).

In Columns A and B, the observation unit is a hospital in our total sample of 23,151 providers in the country from 2011 to 2015. Column A contains results from the bivariate regression of a given hospital quality measure on the CON dummy variable.[9] Column B includes results from a multivariate regression controlling for median income, age, demographics, percentage uninsured, unemployment, and education of people in a

provider's county. In Columns C through F, the observation unit is a hospital in the previously identified subsample of HRRs that contain providers in both CON and non-CON states. Columns C and D contain results for the same bivariate and multivariate regressions as in Columns A and B but consider only the subsample of hospitals in border-crossing HRRs. Column E contains results from a bivariate regression with HRR fixed effects. Column F is our preferred specification and includes results from the HRR fixed-effects model using the restricted sample of hospitals and including the controls from the multivariate regression.

The estimates from the pooled bivariate and multivariate regressions using both the total sample and the restricted sample contained in Table 7, Columns A through D, demonstrate that, in most cases, hospitals in CON states perform worse on the quality indicators than hospitals in non-CON states. Of the statistically significant estimates, 21 out of 23 indicate worse quality in CON states.

In our preferred specification with HRR fixed effects in Table 7, Column F, the estimates of the quality indicators of *Death among Surgical Inpatients with Serious Treatable Complications*, *Pneumonia Mortality Rate*, and *Heart Failure Mortality Rate* are statistically significantly higher in states with CON laws than in non-CON states. By contrast, the Heart Attack Mortality Rate estimate misses significance at the 10% level with a *p*-value of 0.129. Furthermore, all other quality indicators have their predicted sign but are statistically insignificant. The findings support the hypothesis that CON regulations lower the quality of medical services. The change in magnitude of the coefficient in Column F relative to the other columns suggests that unmeasured or unobserved variables are correlated with quality of care and whether a state has a CON law.

Table 7, Column F, shows that the 30-day mortality rate for pneumonia patients is roughly 0.38 percentage points higher in CON states than in non-CON states. Further, the 30-day mortality rate for heart failure is about 0.2 percentage points higher in CON states. This means that the average mortality rate for heart failure in CON states is 1.7 percent higher than the average in non-CON states. The results imply that the average hospital in a state with CON regulations experiences between two and four more deaths per 1000 discharges than hospitals in non-CON states, depending on the illness.

The largest difference for all measures is in *Death among Surgical Inpatients with Serious Treatable Complications*. This measure is a composite of the number of deaths following a severe complication after surgery. The estimate for this measure implies that hospitals in CON states average six more deaths per 1000 surgical discharges that result in complications. The mortality rate from complications is about 5.5 percent higher in CON states than the average mortality rate for the restricted sample.

Table 7, Column F, also shows that readmission rates tend to be either the same or higher in states with CON regulations. However, none of these differences are statistically significant at the five percent level. Furthermore, Column F shows that the difference in the rate of *Postoperative Pulmonary Embolism or Deep Vein Thrombosis* and the percentage of patients giving their hospital an overall HCAHPS rating of 9 or 10 is not significantly different from zero.

One potentially confounding factor that our model does not capture is the impact of the Hospital Readmissions Reduction Program (HRRP), a provision of the Affordable Care Act that penalizes hospitals for excess 30-day readmissions following Medicare fee-for-service patient discharges (CMS (Centers for Medicare & Medicaid Services) 2016b). Penalties are assessed based on hospitals' readmission rates for heart attack, heart failure, and pneumonia. The new provision became applicable to hospital discharges in 2012. Hospitals with higher-than-expected 30-day readmission rates for the three conditions faced a maximum one percent reduction in payments for discharges in 2013, increasing to two percent in 2014 and three percent in 2015.

The penalties associated with the HRRP may account for the absence of systematic differences in readmission rates and those observed for mortality rates. If CON hospitals had higher readmission rates than non-CON hospitals before the HRRP, the penalties

under the program would incentivize those hospitals to lower their readmission rates more quickly than non-CON hospitals. There is some evidence that hospitals are responding to the HRRP. For example, Zuckerman (2016) found that readmission rates fell sharply for the conditions targeted by the HRRP and that they fell less sharply for readmissions following discharges for other hospitalizations. Zuckerman notes that "the drop in readmissions mostly occurred during the enactment of the Affordable Care Act in March 2010 and the start of the Hospital Readmissions Reduction Program in October 2012, when hospitals would have taken action to avoid facing penalties" (Zuckerman 2016, p. 2). This drop-in response to the HRRP coincides with the beginning of our study period. It may partly explain why we do not observe more considerable differences in readmission rates between CON and non-CON hospitals.

Overall, our results do not support the hypothesis that CON hospitals deliver better-quality care than non-CON hospitals. We find the opposite: nearly all the coefficients in our regressions suggest that CON regulations lead to lower-quality care. However, not all estimates are significant in all our specifications.

## 6. Robustness Checks

### 6.1. Regression Results with Health Controls

One concern regarding our estimates in the pooled panel regression model is that unhealthy areas in CON states primarily drive the results. To test whether unhealthy counties drive our results, we test our preferred specification in Table 8 using county population health controls. In column A, we use controls for the percentage of adults that report greater than or equal to a body mass index of 30, i.e., obesity, the percentage of the adults that say they are currently smoking, and percentage of the adults that say they are in fair or poor health (age-adjusted). Once we control for these three factors, our results do not vary significantly from our results in column F of Table 7. The estimates for the effect of CON laws on *Death among Surgical Inpatients with Serious Treatable Complications, Pneumonia Mortality Rate, and Heart Failure Mortality Rate* remain positive and statistically significant.

An alternative way to test whether unhealthy counties drive our results is to separate our dataset into sick and healthy counties: columns B–G test subsamples of the data based on the median of the healthy controls. For example, in column B, we test our preferred specification on counties at, or above, the median obesity measure. This test shows how CON laws affect hospitals in the unhealthiest counties. In column C, we test our preferred specification on the counties falling below the median level of our obesity measure. This test shows CON law effects on hospitals in the healthiest counties. We run the same tests using the percentage of those who report currently smoking and the percentage of adults who say they are in fair health. Columns B–G show that the *Death among Surgical Inpatients with Serious Treatable Complications* is statistically significantly higher in CON states than in non-CON states in healthy and unhealthy counties. The result shows that the effect of CON laws on *Death among Surgical Inpatients with Serious Treatable Complications* is not driven by the current health status of the county. Furthermore, the results show that *Pneumonia Mortality Rate* is statistically significantly higher in CON states than in non-CON states in healthier counties. The Heart Attack Mortality Rate is statistically significantly higher in CON states than in non-CON states in unhealthier counties. Moreover, the Heart Failure Mortality Rate results depend on the health measure used to separate the sample. With our obesity measure, the *Heart Failure Mortality Rate* is statistically significantly higher in CON states than in non-CON states in unhealthier counties. In comparison, the *Heart Failure Mortality Rate* is statistically significantly higher in CON states than in non-CON states in healthier counties using the percentage that reports they are smoking and the percentage that says poor or fair health measures. The results in Table 8 show that the results from the preferred specification are not driven solely by unhealthy counties.

**Table 8.** Pooled regression results (2011–2015).

| Measure Name (CMS Code) | (A) HRR Fixed-Effects Model with Health Controls | (B) HRR Fixed-Effects Model Bottom Fifty Percent Obesity | (C) HRR Fixed-Effects Model Top Fifty Percent Obesity | (D) HRR Fixed-Effects Model Bottom Fifty Percent Smoking | (E) HRR Fixed-Effects Model Top Fifty Percent Smoking | (F) HRR Fixed-Effects Model Bottom Fifty Percent Fair or Poor Health | (G) HRR Fixed-Effects Model Top Fifty Percent Fair or Poor Health |
|---|---|---|---|---|---|---|---|
| Death among Surgical Inpatients with Serious Treatable Complications (deaths per 1000 surgical discharges with complications) (PSI #4) | 5.940 *** (2.128) 1481 | 5.975 * (3.265) 698 | 5.551 * (3.131) 792 | 5.975 * (3.195) 790 | 7.651 *** (2.794) 700 | 7.744 ** (3.221) 598 | 5.149 * (2.762) 892 |
| Postoperative Pulmonary Embolism or Deep Vein Thrombosis (per 1000 surgical discharges) (PSI #12) | 0.0884 (0.155) 2513 | 0.267 (0.193) 1369 | 0.199 (0.139) 1206 | 0.0390 (0.212) 1429 | 0.570 * (0.291) 1146 | 0.0187 (0.191) 1095 | 0.119 (0.253) 1480 |
| Percentage of patients giving their hospital a 9 or 10 overall rating (percentage points) (HCAHPS) | −0.603 (0.964) 3417 | −1.151 (1.139) 1985 | −0.358 (1.111) 1648 | −0.938 (0.997) 1992 | −0.444 (1.599) 1641 | −2.015 * (1.169) 1543 | −0.376 (1.151) 2090 |
| Pneumonia Readmission Rate (percentage points) (READM-30-PN) | 0.0561 (0.106) 3553 | 0.180 (0.133) 2196 | −0.0129 (0.123) 1627 | 0.100 (0.122) 2190 | −0.0673 (0.134) 1633 | 0.00811 (0.172) 1689 | 0.157 (0.0968) 2134 |
| Pneumonia Mortality Rate (percentage points) (MORT-30-PN) | 0.354 *** (0.131) 3526 | 0.319 ** (0.148) 2182 | 0.340 * (0.186) 1617 | 0.236 (0.157) 2183 | 0.595 *** (0.189) 1616 | 0.131 (0.170) 1683 | 0.642 *** (0.162) 2116 |
| Heart Failure Readmission Rate (percentage points) (READM-30-HF) | 0.231 (0.182) 3201 | 0.416 * (0.243) 1959 | 0.195 (0.202) 1468 | 0.386 (0.234) 1990 | −0.0602 (0.216) 1437 | 0.555 * (0.298) 1517 | 0.0936 (0.173) 1910 |
| Heart Failure Mortality Rate (percentage points) (MORT-30-HF) | 0.216 *** (0.0819) 3124 | 0.259 ** (0.130) 1909 | 0.00910 (0.176) 1429 | 0.113 (0.123) 1944 | 0.390 *** (0.130) 1394 | 0.0594 (0.132) 1475 | 0.352 *** (0.104) 1863 |
| Heart Attack Readmission Rate (percentage points) (READM-30-AMI) | 0.144 (0.117) 1714 | −0.0493 (0.218) 885 | 0.360 * (0.204) 862 | 0.173 (0.177) 986 | 0.0557 (0.257) 761 | −0.0156 (0.213) 742 | 0.196 (0.210) 1005 |
| Heart Attack Mortality Rate (percentage points) (MORT-30-AMI) | 0.248 (0.181) 1907 | 0.495 ** (0.219) 1038 | −0.171 (0.238) 918 | 0.385 * (0.199) 1119 | 0.0951 (0.307) 837 | 0.690 *** (0.204) 837 | −0.0412 (0.241) 1119 |
| Controls | Yes | Yes | Yes | Yes | Yes | Yes | Yes |
| Year fixed effets | Yes | Yes | Yes | Yes | Yes | Yes | Yes |
| HRR fixed effects | Yes | Yes | Yes | Yes | Yes | Yes | Yes |

Note: CMS = Centers for Medicare and Medicaid Services; HRR = hospital referral region; HCAHPS = Hospital Consumer Assessment of Healthcare Providers and Systems. The model specifications in Columns A and B consider the full sample of hospitals in the United States. The specifications in Columns C through E consider only hospitals in HRRs that cross the border between CON and non-CON states. The unit of analysis is the individual provider. Clustered standard errors by provider and hospital referral region are in parentheses. Controls for percentage over age 65, percentage African American, percentage Hispanic, percentage rural, average freshman graduation rate (high school), percentage uninsured, median income, and unemployment rate are the average from the county level. Readmission and mortality rates are calculated using data from Medicare patients only. The number of observations varies between the bivariate and multivariate regressions and by measure. In each cell in the table, the top number is the coefficient estimate, the number in parentheses is the standard error, and the bottom number indicates the number of observations. *** Statistically significant at (at least) the 1% level. ** Statistically significant at (at least) the 5% level. * Statistically significant at (at least) the 10% level. Sources: CMS (Centers for Medicare & Medicaid Services) (2016d); Hospital Compare Data Archive (n.d.) (2011, 2012, 2013, 2014, 2015); Dartmouth Atlas Project (2016); American FactFinder (2016); County Health Rankings and Roadmaps (County Health Rankings and Roadmaps 2011–2015).

## 6.2. Regression Results by Year (2011–2015)

To test whether chance findings are driving the pooled panel regression results, we also present the results of the HRR fixed-effects model for the same quality measures for

each year from 2011 to 2015. However, our previous results are unlikely to be driven by changes in the number of HRRs crossing state borders, as that number stayed very similar from 2011 to 2015. Moreover, the number of providers in these border-crossing HRRs remained remarkably static.

The results of these additional tests are mainly consistent with the pooled regression model. Table 9 summarizes these results. *Death among Surgical Inpatients with Serious Treatable Complications* is statistically significantly higher for hospitals in CON states, except in 2013. However, this finding goes with the caveat that only one-third of all hospitals in the restricted sample report this metric.

**Table 9.** Regression results by year (2011–2015).

| Measure Name (CMS Code) | 2011 | 2012 | 2013 | 2014 | 2015 |
|---|---|---|---|---|---|
| Death among Surgical Inpatients with Serious Treatable Complications (deaths per 1000 surgical discharges with complications) (PSI #4) | 10.50 *** (1.441) 292 | 8.152 *** (2.729) 313 | 6.282 (3.954) 305 | 5.235 * (2.589) 290 | 4.168 ** (1.871) 290 |
| Postoperative Pulmonary Embolism or Deep Vein Thrombosis (per 1000 surgical discharges) (PSI #12) | −0.132 (0.260) 503 | 0.0622 (0.154) 516 | 0.134 (0.132) 519 | 0.236 (0.153) 516 | 0.263 (0.190) 521 |
| Percentage of patients giving their hospital a 9 or 10 overall rating (percentage points) (HCAHPS) | −1.289 (1.485) 667 | −1.397 (0.979) 710 | −1.422 (1.042) 730 | −0.624 (0.968) 746 | −0.620 (1.128) 780 |
| Pneumonia Readmission Rate (percentage points) (READM-30-PN) | 0.181 (0.149) 745 | 0.107 (0.119) 800 | 0.0641 (0.0813) 807 | 0.0280 (0.103) 737 | 0.0526 (0.0787) 734 |
| Pneumonia Mortality Rate (percentage points) (MORT-30-PN) | 0.527 *** (0.151) 741 | 0.364 ** (0.137) 795 | 0.319 ** (0.156) 801 | 0.329 *** (0.108) 732 | 0.355 *** (0.125) 730 |
| Heart Failure Readmission Rate (percentage points) (READM-30-HF) | 0.293 (0.240) 687 | 0.214 (0.198) 709 | 0.277 (0.167) 705 | 0.212 * (0.124) 665 | 0.258 * (0.129) 661 |
| Heart Failure Mortality Rate (percentage points) (MORT-30-HF) | 0.319 ** (0.121) 674 | 0.275 *** (0.0822) 690 | 0.128 (0.124) 685 | 0.141 * (0.0792) 647 | 0.213 * (0.124) 642 |
| Heart Attack Readmission Rate (percentage points) (READM-30-AMI) | 0.491 ** (0.194) 352 | 0.484 ** (0.186) 350 | 0.202 (0.134) 351 | −0.0754 (0.0920) 345 | −0.179 (0.149) 349 |
| Heart Attack Mortality Rate (percentage points) (MORT-30-AMI) | 0.338 (0.217) 411 | 0.363 ** (0.140) 403 | 0.173 (0.173) 389 | 0.170 (0.147) 381 | 0.249 (0.188) 372 |
| Number of providers | 921 | 1060 | *1076* | *957* | *999* |

Note: CMS = Centers for Medicare and Medicaid Services; HCAHPS = Hospital Consumer Assessment of Healthcare Providers and Systems. The unit of analysis is the individual provider. Clustered standard errors by state are in parentheses. Controls for percentage over age 65, percentage African American, percentage Hispanic, percentage rural, average freshman graduation rate (high school), percentage uninsured, median income, and unemployment rate are the average from the county level. Readmission and mortality rates are calculated using data from Medicare patients only. *** Statistically significant at (at least) the 1% level. In each cell in the table, the top number is the coefficient estimate, the number in parentheses is the standard error, and the bottom number indicates the number of observations. ** Statistically significant at (at least) the 5% level. * Statistically significant at (at least) the 10% level. Sources: CMS (Centers for Medicare & Medicaid Services) (2016d); Hospital Compare Data Archive (n.d.) (2011, 2012, 2013, 2014, 2015); Dartmouth Atlas Project (2016); American FactFinder (2016); County Health Rankings and Roadmaps (County Health Rankings and Roadmaps 2011–2015).

The difference in *Pneumonia Mortality Rate* between CON and non-CON hospitals is also statistically significant each year, representing between three and five additional deaths per 1000 pneumonia discharges. Furthermore, we find that *Heart Failure Mortality*

*Rate* is higher in CON hospitals than in non-CON hospitals, although the differences are not statistically significant in 2013.

In our regressions by year, we again find that the readmission rates were generally no different at CON hospitals than at non-CON hospitals. However, the estimate for the *Heart Attack Readmission Rate* is different from zero in 2011 and 2012 at the five percent level and has the predicted sign in these two years. In addition, the *Heart Failure Readmission Rate* is different from zero in 2014 and 2015 at the 10 percent level and has the predicted sign in these two years.

Moreover, consistent with our baseline estimates, the difference in the *Postoperative Pulmonary Embolism* rate between CON and non-CON hospitals is not significantly different from zero each year.

We also find a downward trend from 2011 to 2015 in most of our estimates of quality indicators. The implementation of the HRRP may explain the trend discussed earlier. As stated before, penalties were assessed based on hospitals' readmission rates for heart attack, heart failure, and pneumonia. The new provision became applicable to hospital discharges in 2012. Hospitals with higher-than-expected 30-day readmission rates for the three conditions faced a maximum one percent reduction in payments for discharges in 2013, increasing to two percent in 2014 and three percent in 2015.

### 6.3. Regression Results Excluding Low-Provider HRRs

One concern about the previous results might be that results from the pooled panel regression model are sensitive due to the fact that some HRRs in our subsample have only a few hospitals on one side of the state border that runs through them. Table 3 illustrates the potential issue: in 2011, almost one-third of HRRs that crossed the border between CON and non-CON states had only a few hospitals on one side or both sides of the border. If one or more of those hospitals is abnormally high or low performing on the quality indicators, such skewness in the data might drive our findings in Tables 7 and 9.

To address this concern, this study restricts the fixed-effects model to exclude all HRRs with three or fewer providers on one or both sides of the border. We do this for each year from 2011 to 2015.[10] Table 10, Column A, contains the results from our original pooled panel regression model with fixed effects. Column B shows the results for the same model while omitting the HRRs with three or fewer providers on one side or both sides of the border, which we find are largely consistent with the results from Column A.

**Table 10.** Robustness checks (2011–2015).

| Measure Name (CMS Code) | (A) Original Fixed-Effects Model | (B) Omitting Low HRRs | (C) Omitting Unbalanced HRRs | (D) Omitting Low-CON States |
|---|---|---|---|---|
| Death among Surgical Inpatients with Serious Treatable Complications (deaths per 1000 surgical discharges with complications) (PSI #4) | 6.161 *** (2.278) 1490 | 7.38 *** (1.69) 839 | 6.36 *** (2.35) 574 | 6.82 *** (1.87) 1124 |
| Postoperative Pulmonary Embolism or Deep Vein Thrombosis (per 1000 surgical discharges) (PSI #12) | 0.117 (0.150) 2575 | 0.205 (0.221) 1410 | 0.309 (0.278) 1020 | 0.217 (0.140) 1950 |
| Percentage of patients giving their hospital a 9 or 10 overall rating (percentage points) (HCAHPS) | −0.964 (0.957) 3633 | −1.51 (1.051) 2083 | −1.88 ** (0.917) 1411 | −2.55 *** (0.918) 2586 |
| Pneumonia Readmission Rate (percentage points) (READM-30-PN) | 0.0854 (0.0958) 3823 | 0.059 (0.087) 2229 | 0.063 (0.113) 1578 | 0.167 (0.148) 2726 |

**Table 10.** *Cont.*

| Measure Name (CMS Code) | (A) Original Fixed-Effects Model | (B) Omitting Low HRRs | (C) Omitting Unbalanced HRRs | (D) Omitting Low-CON States |
|---|---|---|---|---|
| Pneumonia Mortality Rate (percentage points) (MORT-30-PN) | 0.379 *** (0.122) 3799 | 0.434 *** (0.121) 2216 | 0.410 *** (0.145) 1565 | 0.303 ** (0.149) 2715 |
| Heart Failure Readmission Rate (percentage points) (READM-30-HF) | 0.248 (0.184) 3427 | 0.0457 (0.171) 2021 | 0.0785 (0.193) 1455 | 0.501 ** (0.244) 2450 |
| Heart Failure Mortality Rate (percentage points) (MORT-30-HF) | 0.198 ** (0.082) 3338 | 0.271 *** (0.083) 1955 | 0.210 ** (0.087) 1411 | 0.228 ** (0.101) 2384 |
| Heart Attack Readmission Rate (percentage points) (READM-30-AMI) | 0.179 (0.124) 1747 | 0.132 (0.110) 996 | −0.0447 (0.141) 713 | 0.381 *** (0.138) 1331 |
| Heart Attack Mortality Rate (percentage points) (MORT-30-AMI) | 0.263 (0.173) 1956 | 0.139 (0.215) 1128 | 0.308 (0.239) 808 | 0.212 (0.175) 1476 |

Note: CMS = Centers for Medicare and Medicaid Services; HRRs = hospital referral regions; CON = certificate of need; HCAHPS = Hospital Consumer Assessment of Healthcare Providers and Systems. Column A contains original fixed-effects regression results. Column B contains results after dropping HRRs with three or fewer hospitals on either side of the border. Column C contains results after dropping HRRs that have at minimum four times fewer the number of providers on one side of the border than the other. Column D contains results after dropping observations in states below the median number of CON laws. The unit of analysis is the individual provider. Clustered standard errors by provider and hospital referral region are in parentheses. Controls for percentage over age 65, percentage African American, percentage Hispanic, percentage rural, average freshman graduation rate (high school), percentage uninsured, median income, and unemployment rate are the average from the county level. Controls also include year dummy variables. Readmission and mortality rates are calculated using data from Medicare patients only. In each cell in the table, the top number is the coefficient estimate, the number in parentheses is the standard error, and the bottom number indicates the number of observations. *** Statistically significant at (at least) the 1% level. ** Statistically significant at (at least) the 5% level. Sources: CMS (Centers for Medicare & Medicaid Services) (2016d); Hospital Compare Data Archive (n.d.) (2011, 2012, 2013, 2014, 2015); Dartmouth Atlas Project (2016); American FactFinder (2016); County Health Rankings and Roadmaps (County Health Rankings and Roadmaps 2011–2015).

The magnitudes of the coefficients on CON in the regressions that do not include low-provider HRRs (column B) are similar to the coefficients in the original model (column A). Differences in *Death among Surgical Inpatients with Serious Treatable Complications* and *Pneumonia Mortality Rate*, and *Heart Failure Mortality Rate* between CON and non-CON hospitals, remain statistically significant, and their coefficients increase in magnitude. The measures for readmission rate remain statistically insignificantly different from zero. These results provide evidence that this consideration accounted for the original results in the low-provider HRRs.

*6.4. Regression Results Excluding HRRs with the Most Uneven Number of Hospitals on Each Side of the Border*

Another concern regarding our estimates in the pooled panel regression model is that some of the border-crossing HRRs contain a highly unbalanced number of hospitals on one side of the market compared with the other side. Table 3 again illustrates the potential issue. For instance, in 2011, HRR number 371 contained 46 hospitals on the non-CON side of the border, but only seven on the CON side. To address this potential issue, we further restrict our model to exclude all HRRs in which there are at least four times more providers on one side of the border than the other. This omits 23 HRRs and 2877 providers from our subsample. Table 10, Column C, contains the pooled panel regression results missing these unbalanced HRRs, and we find the results are very similar to those in Columns A and B, with the exception that *the Percentage of patients giving their hospital a 9 or 10 overall rating* becomes statistically significant and has its predicted sign.

*6.5. Regression Results Excluding States with Few CON Laws*

In our original model, a state is defined as a CON state with at least one CON regulation. However, the effects of CON regulations may be cumulative, meaning that states with many entry restrictions may see more considerable quality differences than states with relatively few. In this case, we would expect states with only a few CON laws to look more like non-CON states in terms of hospital quality than states with more comprehensive CON programs. By treating all CON states the same, our model could miss these cumulative effects and thus underestimate the true impact of CON laws on hospital quality.

To address this issue, we further restrict our subsample to exclude hospitals in any CON state with fewer than four CON laws, the median number of laws for the CON states in our subsample. This omits 1364 providers and 10 HRRs from the subsample. The results are again consistent with the original pooled regression model and provide evidence that states with the most restrictive CON programs have lower-quality hospitals than non-CON states.

Table 10, Column D, contains the pooled panel regression results omitting states with the fewest CON laws. As in the original model, differences between CON and non-CON hospitals in *Death among Surgical Inpatients with Serious Complications*, *Pneumonia Mortality Rate*, and *Heart Failure Mortality Rate* remain statistically significant. Furthermore, estimates for the difference in *Percentage of patients giving their hospital a 9 or 10 overall rating*, *Heart Failure Readmission Rate*, and *Heart Attack Readmission Rate* are also statistically significant using the restricted sample. Using this restricted sample, we find that CON hospitals have, on average, two-and-a-half percentage points fewer patients rating their hospital a 9 or 10 overall on the HCAHPS survey than non-CON hospitals.

*6.6. Aggregate Hospital Quality Measures*

One possible limitation of our previous findings may be that our quality variables do not fully capture all aspects of provider quality. This limitation stems from two issues: The first is that there is no consensus about the most important individual variables to examine when assessing overall hospital quality. The second is that no aggregate measures were designed to allow for high-level comparisons across hospitals. This section attempts to compensate for the second issue by incorporating five additional quality measures meant to capture hospital quality at a more aggregate level.

Goodman et al. (2011) used data on Medicare patients to construct five hospital-level metrics that capture the quality of care for patients who have had medical and surgical procedures. The first post-discharge event is the *30-Day Readmission Rate after Medical Discharge*, which captures readmissions within 30 days of the discharge as a percentage of all Medicare patients classified as having a "medical" hospital visit.[11] The second event is *14-Day Ambulatory Visit Rate after Medical Discharge*, which measures the percentage of medical patients who require outpatient care within 14 days of discharge. The third event is the *30-Day Emergency Room Visit Rate after Medical Discharge*, which measures the percentage of medical patients who visited the emergency room within 30 days of discharge. The final two events are the *30-Day Readmission Rate after Surgical Discharge* and *30-Day Emergency Room Visit Rate after Surgical Discharge*, which capture the percentage of Medicare patients who underwent a "surgical" procedure and were readmitted within 30 days of discharge and the percentage that visited the emergency room within 30 days of release, respectively.

Hospital-level data for those five indicators are available from the Dartmouth Atlas Project from 2011 to 2013. The data were collected from CMS's Medicare Provider Analysis and Review File. Patients included in the case mix were Medicare fee-for-service beneficiaries with full Medicare Part A and Part B coverage during the study period. Patients who left against medical advice were discharged to hospice care or died while in the hospital and were excluded from the sample. The rates were adjusted for age, gender, and race. See Goodman et al. (2011) for more detail about how this measure was constructed.

We also analyzed a second set of indicators in 2013, when CMS began calculating several composite quality measures meant to serve as better hospital performance indicators

across a class of metrics. These include an all-cause hospital readmissions rate and a composite rate of complications after surgery. Hospital-level data for these two indicators were available from Hospital Compare for 2013–2015.

The *Hospital-Wide 30-Day Readmission Rate* (READM-30-HOSP-WIDE) is a summary rate of unplanned readmissions within 30 days of discharge for all medical, surgical, cardiorespiratory, cardiovascular, and neurological conditions and procedures. According to Rosen et al. (2016), these five patient cohorts represent almost 90 percent of hospital admissions. Patients included are from the Medicare fee-for-service population age 65 and older who were discharged from any nonfederal, short-stay, acute-care hospital, or critical access hospital (Horwitz et al. 2011). Like the other CMS readmission and mortality rates used in this study, the all-cause readmission rate is risk-standardized to consider an individual hospital's case mix. The all-causes readmission rate also adjusts for each hospital's patients' primary diagnosis to consider variations in conditions and procedures, allowing for comparison across heterogeneous providers.

The *Aggregate Patient Safety Indicator* (PSI #90) captures how well a hospital prevents complications after surgery compared to other hospitals with a similar case mix. This measure is a weighted average of the hospital's performance on the following complications: pressure ulcer, iatrogenic pneumothorax, central venous catheter-related bloodstream infection, postoperative hip fracture, postoperative hemorrhage or hematoma, postoperative physiologic and metabolic derangement, postoperative respiratory failure, postoperative pulmonary embolism or deep vein thrombosis, postoperative sepsis, postoperative wound dehiscence, and accidental puncture or laceration (note that the composite measure does not include deaths from severe complications after surgery). The resulting composite ratio is scaled to an expected score of one, given a hospital's case mix. A score of more than one indicates that the hospital had more complications than other hospitals with a similar case mix. In contrast, a score of less than one indicates fewer complications than hospitals with a similar case mix. For more detail about how this measure is constructed, see AHRQ (Agency for Healthcare Research and Quality) (2010).[12]

Table 11 contains the summary statistics of these measures. Panel A compares these aggregate quality measures in CON and non-CON states, and Panel B compares the indicators at hospitals in the subsample of HRRs that cross the border between CON and non-CON states. As in the previous robustness checks, the results of the pooled regression model with fixed effects when using these aggregate quality measures are broadly consistent with our original model. We generally find that hospitals in CON states perform either worse or the same as non-CON hospitals on these additional quality measures. However, not all differences are statistically significant at conventional levels.

**Table 11.** Difference-in-means tests: aggregate quality measures.

| Panel A: All CON States versus All Non-CON States | Non-CON States | CON States | Difference | Clustered *t* Statistic | Observations |
|---|---|---|---|---|---|
| 30-Day Readmission Rate after Medical Discharge (percentage points) | 15.0 | 15.5 | −0.5 | 7.05 | 9341 |
| 14-Day Ambulatory Visit Rate after Medical Discharge (percentage points) | 63.8 | 64.2 | −0.4 | 1.32 | 11,811 |
| 30-Day Emergency Room Visit Rate after Medical Discharge (percentage points) | 19.3 | 20.1 | −0.9 | 9.26 | 10,163 |
| 30-Day Readmission Rate after Surgical Discharge (percentage points) | 11.2 | 12.0 | −0.8 | 6.41 | 5387 |

**Table 11.** *Cont.*

| Panel A: All CON States versus All Non-CON States | Non-CON States | CON States | Difference | Clustered *t* Statistic | Observations |
|---|---|---|---|---|---|
| 30-Day Emergency Room Visit Rate after Surgical Discharge (percentage points) | 15.0 | 15.8 | −0.8 | 6.69 | 6150 |
| Hospital-wide 30-Day Readmission Rate (percentage points) | 15.4 | 15.7 | −0.3 | 13.42 | 13,235 |
| Aggregate Patient Safety Indicator (ratio) | 0.75 | 0.75 | 0.0 | −0.51 | 9815 |
| Panel B: HRRs in Both CON and Non-CON States | HRRs in Non-CON States | HRRs in CON States | Difference | Clustered *t* Statistic | Observations |
| 30-Day Readmission Rate after Medical Discharge (percentage points) | 15.1 | 15.4 | −0.3 | 1.67 | 1600 |
| 14-Day Ambulatory Visit Rate after Medical Discharge (percentage points) | 62.2 | 63.8 | −1.7 | 2.02 | 2215 |
| 30-Day Emergency Room Visit Rate after Medical Discharge (percentage points) | 19.3 | 19.9 | −0.6 | 2.78 | 1774 |
| 30-Day Readmission Rate after Surgical Discharge (percentage points) | 11.2 | 11.5 | −0.2 | 0.74 | 877 |
| 30-Day Emergency Room Visit Rate after Surgical Discharge (percentage points) | 14.9 | 15.4 | −0.5 | 1.62 | 988 |
| Hospital-wide 30-Day Readmission Rate (percentage points) | 15.4 | 15.6 | −0.1 | 3.01 | 2735 |
| Aggregate Patient Safety Indicator (ratio) | 0.77 | 0.77 | 0.0 | −1.53 | 1690 |

Note: CON = certificate of need; HRRs = hospital referral region. Rates of readmissions, ambulatory visits, and emergency room visits are from the Dartmouth Atlas of Health Care for 2011–2013 (Dartmouth Atlas Project 2016). The hospital-wide readmission rate and aggregate patient safety indicators are from Hospital Compare for 2013, 2014, and 2015 (Hospital Compare Data Archive (n.d.) 2013, 2014, 2015). The unit of analysis is the individual hospital. Data are collected at the individual hospital level. All rates except the aggregate patient safety indicator are calculated using data from Medicare patients only. All *t* statistics are clustered at the individual provider level. Sources: CMS (Centers for Medicare & Medicaid Services) (2016d); Hospital Compare Data Archive (n.d.) (2013, 2014, 2015); Dartmouth Atlas Project (2016); American FactFinder (2016).

Table 12 contains the results for the pooled panel regression with HRR fixed effects using these new aggregate quality indicators. We find that the *30-Day Readmission Rate after Surgical Discharge* and the *30-Day Emergency Room Visit Rate after Surgical Discharge* are 1.02 and 1.06 percentage points higher at CON hospitals. These estimates are statistically significant at the one percent level. The differences in the *30-Day Readmission Rate after Medical Discharge* and the *14-Day Ambulatory Visit Rate after Medical Discharge* are not significantly different from zero. Similarly, the *30-Day Emergency Room Visit Rate after Medical Discharge*, the *Hospital-wide 30-Day Readmission Rate*, and the *Aggregate Patient Safety Indicator* are not significantly different from zero.

**Table 12.** Regression results for aggregate quality measures.

| Measure Name | Coefficient on CON |
|---|---|
| 30-Day Readmission Rate after Medical Discharge, 2011–2013 (percentage points) | 0.180 (0.174) 1553 |
| 14-Day Ambulatory Visit Rate after Medical Discharge, 2011–2013 (percentage points) | 0.070 (1.39) 2143 |
| 30-Day Emergency Room Visit Rate after Medical Discharge, 2011–2013 (percentage points) | 0.402 (0.316) 1721 |
| 30-Day Readmission Rate after Surgical Discharge, 2011–2013 (percentage points) | 1.02 *** (0.305) 860 |
| 30-Day Emergency Room Visit Rate after Surgical Discharge, 2011–2013 (percentage points) | 1.06 *** (0.464) 967 |
| Hospital-wide 30-Day Readmission Rate, 2013–2015 (percentage points) | 0.071 (0.056) 2522 |
| Aggregate Patient Safety Indicator, 2013–2015 (ratio) | 0.009 (0.021) 1632 |

Note: CON = certificate of need. Rates on readmissions, ambulatory visits, and emergency room visits are from the Dartmouth Atlas of Health Care for 2011–2013 (Dartmouth Atlas Project 2016). The hospital-wide readmission rate and aggregate patient safety indicators are from Hospital Compare for 2013–2015 (Hospital Compare Data Archive (n.d.) 2013, 2014, 2015). The unit of analysis is the individual provider. The unit of analysis is the individual provider. Clustered standard errors by provider and hospital referral region are in parentheses. Controls for percentage over age 65, percentage African American, percentage Hispanic, percentage rural, average freshman graduation rate (high school), percentage uninsured, median income, and unemployment rate are the average from the county level. Controls also include year dummy variables. All rates except the aggregate patient safety indicator are calculated using data from Medicare patients only. *** Statistically significant at (at least) the 1% level. Sources: CMS (Centers for Medicare & Medicaid Services) (2016d); Hospital Compare Data Archive (n.d.) (2013, 2014, 2015); Dartmouth Atlas Project (2016); American FactFinder (2016).

## 7. Conclusions

As of 2016, 36 states and the District of Columbia have some form of regulation requiring healthcare providers to demonstrate a need for their medical services before building new facilities, expanding existing facilities, or offering new procedures.

Theoretically, the effect of CON regulations on the quality of healthcare supplied by providers is ambiguous. Supporters claim that CON laws increase equilibrium quality by restricting the number of providers and ensuring that each provider treats a higher volume of patients than the provider otherwise would, making such providers more proficient. Opponents of CON regulations argue that healthcare providers, as with providers of other goods and services, compete on different margins and that quality of care is one margin. Thus, by artificially restricting the number of providers in a market, CON laws reduce the competitive pressures for incumbent providers, which in turn results in lower-quality services.

Empirical research on the effect of CON laws on healthcare quality generally finds no significant differences between providers in states with and without these regulations. However, most of these studies suffer from two drawbacks: they lack a measure that captures the overall quality of a hospital's medical services and they are unable to isolate the causal effect of CON laws on hospital quality.

We developed an empirical framework that allows us to estimate the effect of the presence of CON laws on the quality of hospitals. Analyzing nine quality indicators and estimating the impact of CON laws based only on how hospital quality varies within the

same healthcare market, we found no evidence that CON laws increase quality of care. Instead, we found evidence consistent with the hypothesis that limiting entry results in lower hospital quality.

For example, we found that mortality rates are statistically significantly higher at hospitals in CON states than in non-CON states. Our findings show that the estimated average 30-day mortality rate for patients discharged with pneumonia and heart failure from hospitals in CON states is between 1.7 and 3.2 percent higher than the average mortality rate for all hospitals in our subsample of HRRs that contains providers in both CON and non-CON states, depending on the illness. We also found that hospitals in CON states average six more deaths per 1000 surgical discharges that result in complications. These findings are mainly robust to various alternative samples and quality measures.

These findings suggest that CON laws are harmful to patients. The results in this study and other studies (see, for example, Bailey 2018, 2021; Baker and Stratmann 2021; Chiu 2021; Mitchell and Stratmann 2022) indicate that CON laws do not have the beneficial effects as they are intended for. Future work on whether CON laws are particularly harmful to vulnerable populations will further inform policy akers about the presence of any unintended consequences of CON laws.

One limitation of the approach used in this article is that not all hospitals report their quality metrics. The higher the reporting rate, the stronger confidence that selection effects do not drive the findings. Additionally, while the research design in the regressions based on the pooled sample accounts for time-invariant omitted variables, one might be concerned about the presence of omitted variables that vary over the five-year span analyzed in these regression models. To the extent that omitted variables change over time and are also correlated with both CON laws and the quality measures, the estimates suffer from an omitted variable bias. Another caveat is the possibility that hospitals on the CON side of a border may compete with hospitals on the non-CON side within each of our border-crossing healthcare markets. Hospitals in CON states might improve the quality of their care due to competition from potentially higher-quality hospitals in non-CON states. Despite this caveat, our approach still finds a quality differential. However, hospitals in CON states outside HRR market areas may provide even worse quality than hospitals in CON states competing with hospitals in non-CON states in the same market. Future research might explore this aspect of non-price competition.

**Funding:** This research received no external funding.

**Institutional Review Board Statement:** Not applicable.

**Informed Consent Statement:** Not applicable.

**Data Availability Statement:** Data available.

**Acknowledgments:** I would like to thank David Wille, the co-author of a working paper from which this article originates. And thank you to Colin Doran for valuable research assistance.

**Conflicts of Interest:** The author declares no conflict of interest.

## Notes

[1] The National Health Planning and Resources Development Act of 1974, the original impetus for CON laws, contains the following language in its statement of purpose: "The massive infusion of Federal funds into the existing health care system has contributed to inflationary increases in the cost of health care and failed to produce an adequate supply or distribution of health resources, and consequently has not made possible equal access for everyone to such resources". Pub. L. No. 93-641 (1975).

[2] The empirical design proposed here is similar in spirit to the design employed by Chiu (2021).

[3] For a summary of Virginia's application process, see Virginia Department of Health (2015, p. 18).

[4] A hospital's performance on the *Deaths among Surgical Inpatients with Serious Treatable Complications* measure is an accurate indicator of quality of care, assuming providers in CON and non-CON states turn away patients at the same rate. If this assumption does not hold, it may be that hospitals in CON states only appear to perform worse on this measure. For example, if CON regulations give incumbents the market power to be able to turn away all but the most seriously ill patients, the CON hospitals' quality metrics would tend to be lower because they are treating a pool of less healthy patients, not because they



provide lower-quality care. Alternatively, use of the *Deaths among Surgical Inpatients with Serious Treatable Complications* measure will result in underestimation of the effect of CON laws on hospital quality if patients with the most serious risk of dying choose high-quality hospitals and if those patients develop complications not because of poorer hospital care but because they are very ill. Therefore, the direction of the potential bias is theoretically ambiguous.

[5]     For more detail about the patient mix adjustment, see CMS (Centers for Medicare & Medicaid Services) (2008b).

[6]     For more detail about how these measures are calculated, see QualityNet (2016).

[7]     This research studies whether the existence of at least one CON law in a state influences the quality of hospital care. An alternative design might study the effect of those CON laws that directly affect hospital services.

[8]     One interesting feature of the quality data is that, on average, the quality of medical services is increasing over time. All regression specifications for the pooled sample include year-fixed effects.

[9]     We use year indicators in all columns.

[10]    For 2011, this criterion eliminates 24 HRRs and 417 providers from our subsample. For 2012, we exclude 23 HRRs and 414 hospitals. For 2013, we exclude 23 HRRs and 419 hospitals. For 2014, we exclude 21 HRRs and 401 hospitals. For 2015, we exclude 22 HRRs and 427 hospitals.

[11]    For a list of conditions and procedures categorized as "medical" and "surgical", see CMS (Centers for Medicare & Medicaid Services) (2008a).

[12]    Some of the measures by AHRQ might suffer from the problem that the individual measures from which the published aggregate measures are constructed are only few.

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
