# Peer review of "The Effects of Certificate-of-Need Laws on the Quality of Hospital Medical Services"

_jrfm, doi:10.3390/jrfm15060272_

Round 1

Reviewer 1 Report

Certificate-of-Need Laws and Hospital Quality

Major Revision

Major concerns

Review post-2016 literature—a quick search in google scholar reveals quite a few relevant works post 2018

Review criteria for indicating CON with respect to hospital quality.  None of the OH, LA for example CON regulations relate to hospital, you may want to limit CON to hospital related regulations

Table of means could show progression over time in quality indicator rates—many are declining over the periods as the spotlight is on quality and policy is enacted (does mention Hospital Readmission Reduction program), CMS Partnership for patients

Are there time fixed effects in the model?

The main significant finding has very low reporting (Death among Surgical Inpatients with Serious Treatable Complications), less than a third of hospitals in the restricted sample—this should be discussed with the results and conclusions

Need to include a discussion of the “small numbers problem” for some of these measures which make the composite AHRQ measure which are s=insignificant when tests perhaps more appropriate

Is there control for hospital ownership, size and teaching status (all associate with errors and may vary by CON states within an HRR)

In the table, 8-11 indicate in the table notes that the third number reported is the sample size, or if not what it is.

Minor-

Careful editing required.  There are sentence fragments, incomplete words and the like.

Author Response

Thank you very much for your careful review of my manuscript and for your constructive and helpful suggestions which resulted in an improvement of my paper.

 Reviewer:

Review post-2016 literature—a quick search in google scholar reveals quite a few relevant works post 2018

Response:

Thank you for alerting me to this issue. I have now cited an additional 17 relevant papers and articles, all of which have been published after 2016.

Reviewer:

Review criteria for indicating CON with respect to hospital quality.  None of the OH, LA for example CON regulations relate to hospital, you may want to limit CON to hospital related regulations

Response:

Thank you for pointing this out. In response, I have added a footnote stating “This research studies whether the existence of at least one CON law in a state influences the quality of hospital care. An alternative design might study the effect of those CON laws that directly affect hospital services.”

Reviewer:

Table of means could show progression over time in quality indicator rates—many are declining over the periods as the spotlight is on quality and policy is enacted (does mention Hospital Readmission Reduction program), CMS Partnership for patients

Are there time fixed effects in the model?

Response:

To save space on an already lengthy paper, I did not add to the table, showing the progression of improvements in quality indicators over time. However, I double-checked the quality data, found the pattern you mentioned, and added a footnote stating “One interesting feature of the quality data is that, on average, quality of medical services is increasing over time.” I also noted the inclusion of year fixed effects in this footnote and each regression table that uses the pooled sample as a row indicating that the specification includes year fixed effects.

 Reviewer:

The main significant finding has very low reporting (Death among Surgical Inpatients with Serious Treatable Complications), less than a third of hospitals in the restricted sample—this should be discussed with the results and conclusions.

Response:

Thank you for pointing out the low reporting. I have now included a discussion of this feature of the data in the sections of the paper with the results and the section of the paper with the conclusion.

Reviewer:

Need to include a discussion of the “small numbers problem” for some of these measures which make the composite AHRQ measure which are insignificant when tests perhaps more appropriate

Response:

Thank you for pointing this out. In this revised version of the paper, I have noted this issue and concern by adding a footnote.

Reviewer:

Is there control for hospital ownership, size and teaching status (all associate with errors and may vary by CON states within an HRR)

Response:

I did not include these variables because the regression model includes HRR fixed effects. So variables that do not change over time within an HRR are controlled for by the HRR fixed effect. Since I study a relatively short period (five years), it seems reasonable to assume that ownership, size, and teaching status do not change over this time span

 Reviewer:

In the table, 8-11 indicate in the table notes that the third number reported is the sample size, or if not what it is.

Response:

I have implemented your recommendation. Thank you for pointing this out. The footnote to each of these tables indicates that the third number reported is the sample size.

Reviewer:

Minor - Careful editing required.  There are sentence fragments, incomplete words, and the like.

Response:

Thank you for pointing out these issues. I carefully edited the paper and I believe these issues are addressed in the current version of this manuscript.

Reviewer 2 Report

Certificate of Need Laws and Hospital Quality

The study examines the causal linked between Certificate of Need (C0N) law and hospital quality for 36 states, using pooled regression analysis and reported hospital service quality has a significant margin on which hospital compete. The result confirmed that CON regulations lead to lower quality care for some quality measures.

Overall, the author lauded for his work. However, the following are my thoughts on how to improve the paper.

 The abstract needs to be remotivated to indicates why Certificate of Need (C0N) laws and hospital quality matters for the States examined.  

The introduction is a bit weak, and paragraph 1-5 contains categorical statement that are cited. The author needs some literature to back up their claims.

The data section is fine and well-motivated.

The author claimed that previous research finds no significant differences on the effects of CON laws on health care quality for providers in states with and without regulations. However, the present research also suffers from serious shortcoming as omitted variable bias has not been accounted for.   

From the results CON laws appears that have different marginal impact on the hospital quality indicators which has not been explored by the author.

The conclusion seems fine, but the policy implications of the study is omitted. Similarly, the suggestion for future research and limitation of the study is not included.

The author needs to address this issue before the paper can be published.

Author Response

Thank you very much for your careful review of my manuscript and for your constructive and helpful suggestions which resulted in an improvement of my paper. 

Reviewer

The study examines the causal linked between Certificate of Need (C0N) law and hospital quality for 36 states, using pooled regression analysis and reported hospital service quality has a significant margin on which hospital compete. The result confirmed that CON regulations lead to lower quality care for some quality measures.

Overall, the author lauded for his work. However, the following are my thoughts on how to improve the paper.

The abstract needs to be remotivated to indicates why Certificate of Need (C0N) laws and hospital quality matters for the States examined.  

Response:

Thank you for this suggestion. I have modified the abstract providing a better motivation. 

Reviewer:

The introduction is a bit weak, and paragraph 1-5 contains categorical statement that are cited. The author needs some literature to back up their claims.

I have now cited literature supporting any claims. I did not cite predictions about quality and prices in the face of more or less competition, as this material is mainstream economics, covered in introductory economics courses. I  hope you will find this acceptable.

Reviewer:

The data section is fine and well-motivated.

The author claimed that previous research finds no significant differences on the effects of CON laws on health care quality for providers in states with and without regulations. However, the present research also suffers from serious shortcoming as omitted variable bias has not been accounted for.   

Thank you for raising this point. I have added a few sentences in the conclusion where I discuss the limitations of this study. I note that to the extent that there are omitted variables that change over time and are correlated with having a CON law and the quality measures, the estimates are suffering from an omitted variable bias.

Reviewer:

From the results CON laws appears that have different marginal impact on the hospital quality indicators which has not been explored by the author.

Thank you for pointing out that the marginal impact differs across measures and that this has not been explored in this paper. The reason for not exploring this feature of the results is detailed micro-level data are not available to explore these issues. Moreover, it may be that some illnesses are inherently more difficult to treat than others, resulting in differences in the marginal impact.

Reviewer:

The conclusion seems fine, but the policy implications of the study is omitted. Similarly, the suggestion for future research and limitation of the study is not included.

I added policy implications to the abstract and suggested future research to the conclusion. The last paragraph of the conclusion now includes limitations and caveats.

Reviewer:

The author needs to address this issue before the paper can be published.

Response

Thank you very much for your constructive comments, which have improved this paper.

Round 2

Reviewer 1 Report

CON Review--Fix up the the language--it looks like you omitted parts of these sentences

Besides quality, another relevant margin on which hospitals compete is the price of medical services. However, hospital reimbursement though insurers, tends to be administratively determined, rather than through market forces, implying that hospitals have a limited in competing through pricing and therefore have an incentive to compete more intensely on non-price margins such as the quality of medical services (Li and Dor 2015) p.2

Economic theory predicts that free entry and competition when regulators among hospitals face regulated prices will increase the equilibrium quality of patient care. P.2

In states with CON programs, healthcare providers seeking to enter a mat, expand their facilities, or offer new services must apply to their state’s healthcare planning agency for approval P.4

Reviewer 2 Report

Dear Author, 

Thank you for sending the revised version of your manuscript. 

I think the paper can be accepted in this present form. 

Best, 

Reviewer